# ExDBSCAN: Explaining DBSCAN with Counterfactual Reasoning

## Abstract

Clustering is an unsupervised technique for grouping data points by similarity. While explainability methods exist for supervised machine learning, they are not directly applicable to clustering, making it challenging to understand cluster assignments. This interpretability gap is evident in the popular density-based method DBSCAN, which assigns points as inliers (cluster members in dense regions) or outliers (noise points in sparse regions). DBSCAN does not provide insight into why a particular point receives its assignment or if it is robust to small changes in the data. To address the challenges, we introduce ExDBSCAN, a density-aware, post-hoc explanation method. ExDBSCAN offers actionable counterfactual explanations, with theoretical guarantees for validity. It generates multiple counterfactuals using a density-connected weighted graph while adopting a physics-inspired model that repels counterfactual candidates from one another (diversity) while pulling them toward the instance to explain (proximity). Empirical evaluation on 30 tabular datasets, confirms that ExDBSCAN attains perfect validity and shows that it retrieves diverse, proximal counterfactuals.

## 1 Introduction

Clustering is a cornerstone of unsupervised machine learning, enabling discovery of patterns and groupings in data without relying on predefined labels. Our work focuses on Density-Based Spatial Clustering of Applications with Noise (DBSCAN) (Ester et al., 1996), an algorithm known for its ability to find clusters of arbitrary shape, and to identify outliers (also denoted as noise) in the process. DBSCAN is popular across domains ranging from computer vision (Figueiredo & Mendes, 2024), to bioinformatics (Francis et al., 2011), and fraud detection (Luo et al., 2024). Nevertheless, a limitation of DBSCAN (and generally clustering methods) is their lack of explanation, i.e., absence of information on why a particular point is belongs to a particular cluster or not, which is crucial for interpretability, user trust, and actionable insights (Miller, 2019; Molnar, 2020).

In supervised learning, counterfactual explanations (CEs) emerge as an impactful tool, identify minimal changes to an input that would alter the model's prediction, offering intuitive "what-if" scenarios (Guidotti, 2024; Poyiadzi et al., 2020; Karimi et al., 2020; Mothilal et al., 2020). Counterfactual reasoning builds user trust in the model and reflects how human stakeholders formulate explanations, making counterfactuals easy to interpret (Miller, 2019). For classification tasks, a typical CE shows how a rejected loan application could be altered to become approved.

We claim that CEs for clustering bring important benefits. Beyond explaining individual points, multiple counterfactuals can reveal alternative ways in which a point could transition between clusters, offering complementary perspectives on the underlying structure. Furthermore, analysing how the set of counterfactual changes across several points enables the possibility of highlighting which features are most influential for cluster membership. For instance, we can identify if outliers systematically need to increase on a particular feature to become inliers, or if points from one cluster need to change a specific attribute to belong to another. These patterns not only enhance interpretability but also enable practitioners to make informed decisions about data preprocessing, feature engineering, and outlier treatment based on a deeper understanding of the clustering structure.

Devising CEs for DBSCAN presents major challenges. First, the cluster assignment process is not based on a differentiable loss function, preventing the counterfactual method from having access to model gradients, often used to guide counterfactual search (Mothilal et al., 2020; Wachter et al.,

2017). Moreover, while classification algorithms often produce class probabilities, a membership probability, used by several model-agnostic counterfactual methods (Mothilal et al., 2020; Yang et al., 2022), is difficult to define in hard clustering contexts. Furthermore, DBSCAN can label points as noise. Noise points can be far from each other, scattered across the whole feature space and share few similarities, having a boundary very different from classes in a classification scenario.

Model-agnostic Bayesian and genetic counterfactual methods for classification, do not generally need access to gradients or class probabilities. Their application though, is generally computationally expensive (Romashov et al., 2022). Furthermore, different clustering algorithms rely on different concepts of similarity. This can impact counterfactual search for example when ensuring diversity, i.e., making sure that different counterfactuals for the same input are not repetitive but instead add meaningful actionable alternatives. In the DBSCAN case, two samples could be close in the traditional Euclidean distance leveraged by most model-agnostic methods; at the same time, they could be only connected by a long density-connected path making the two points effectively distant.

In this work, we address these challenges by leveraging counterfactual reasoning to explain density-based clustering. We propose **Ex**plaining **DBSCAN** with counterfactual reasoning (ExDBSCAN), the first method tailored to generate CEs for DBSCAN. Given a specific point, DBSCAN's assignments and its parameter settings, ExDBSCAN generates counterfactuals that are both proximal (close to the original instance for realistic changes) and diverse (providing multiple non-redundant alternatives), while accounting for DBSCAN's similarity definition. To achieve this, ExDBSCAN models the problem as a physical optimisation system, as it defines an undirected weighted graph that captures DBSCAN's density-based similarity structure. The system treats candidate counterfactuals as charged-like particles that repel each other to ensure diversity, and uses spring-like forces to maintain proximity to the explained instance. Our approach provably provides perfect validity, i.e., every counterfactual attains the correct cluster assignment, for both noise-to-cluster and cluster-to-cluster transitions. Furthermore, ExDBSCAN respects feature constraints by handling non-actionable attributes, e.g., the number of previous hospital visits in a tabular dataset of clinical analysis, changing the non-actionable attributes can make the explanation unrealistic; ensuring the explanations remain practical and applicable. Our contributions are summarised as follows:

- We introduce ExDBSCAN, the first method to generate counterfactual explanations designed for DBSCAN, covering both noise-to-cluster and cluster-to-cluster transitions.
- ExDBSCAN generates diverse counterfactuals under our novel physics-inspired model that repels candidate counterfactuals from one another (diversity) while pulling them toward the point to explain (proximity), all while accounting for cluster structure captured in our graph representation.
- On 30 OpenML tabular datasets, ExDBSCAN outperforms a model-agnostic Bayesian baseline (BayCon) by obtaining lower proximity, higher multi-counterfactual diversity and perfect validity.

## 2  RELATED WORK

In the following related work we review supervised, model-agnostic counterfactuals; explainable clustering; and counterfactuals for clustering and outliers.

**Supervised Model-Agnostic Counterfactual Explanations.** In supervised learning, Wachter et al. (2017) formally introduce CEs as answers to "what is the smallest change to input $x$ that would alter the model's prediction?", and subsequent work optimises criteria such as *proximity*, *validity*, *diversity*, and *actionability* (Guidotti, 2024; Mehdiyev et al., 2024; Yuan et al., 2024).

For model-agnostic methods, e.g. DiCE returns a set of diverse CEs (Mothilal et al., 2020). FACE follows data-supported feasible paths on the empirical manifold and returns a plausible path together with a counterfactual endpoint that alters model prediction (Poyiadzi et al., 2020). MACE uses a reinforcement-learning policy and returns one or more counterfactuals with minimal changes (Yang et al., 2022). All methods need to repeatedly query a supervised model for class probabilities at candidate points and then choose based on closeness to the target class. By contrast, DBSCAN provides only discrete, connectivity-based memberships (core, border, noise) defined by $\varepsilon$ and $minPts$, with no continuous quantity to optimise against. A workaround could be to fit a surrogate classifier to predict cluster memberships and then apply supervised CE methods; however, the surrogate's decision regions would then reflect its own similarity notion rather than density-reachability.

To accurately compare ExDBSCAN with competitive methods, the method Model-Agnostic Bayesian Counterfactual Generator (BayCon) (Romashov et al., 2022) offers a more suitable and adaptable approach for clustering scenarios and thereby is a valid competitor. Unlike gradient-based methods, BayCon uses Bayesian optimisation with probabilistic feature sampling to explore the feature space without requiring differentiable model outputs. It treats the clustering algorithm as a black box, repeatedly querying it with candidate counterfactuals and using the discrete cluster assignments to guide its search. This sampling-based strategy aligns naturally with DBSCAN's discrete membership criteria. However, it does not exploit DBSCAN's specific density-connectivity structure that ExDBSCAN leverages for efficient and theoretically grounded counterfactual generation.

**Explainable Clustering Analysis.** Explainable clustering methods span both global and local explainability. Global approaches aim to summarise entire clusterings or cluster characteristics in human-understandable forms. Moshkovitz et al. (2020) introduce explainable $k$-means/$k$-medians that replace clusters with compact decision trees. Similarly, Bandyapadhyay et al. (2023) formalise searching for small trees that closely match a given partition. Prototype-based methods characterise each cluster by representative examples or "criticism" points, e.g., Kim et al. (2016) propose MMD-Critic to select prototypical instances and outliers for each cluster. Such global explanations convey what defines each cluster overall, but they do not explain individual assignments.

In contrast, local explainability focuses on specific data points. Agnostic attribution frameworks like FACT (Scholbeck et al., 2023) quantify each feature's influence on a points cluster assignment by perturbing features and observing changes in the clustering. Interactive tools like ClusterVision (Kwon et al., 2018), let users vary clustering parameters and visualise effects on clusters and outliers, supporting sensitivity analysis and comparative explanation.

While global summaries, prototypes, feature attributions, and visual analytics improve understanding of clustering results, they do not provide actionable local changes for a given point. No existing method can suggest how a particular DBSCAN inlier or outlier could change its status through minimal feature changes, with guarantees of valid membership under DBSCAN's density-reachability. This gap motivates ExDBSCAN, and focuses on local actionability which generates concrete perturbations to an individual point guaranteeing a change in its DBSCAN assignment.

**Counterfactuals for Clustering or Outlier Detection.** Some recent work studies CEs for clustering. For prototype/mixture-style clustering, Vardakas et al. (2025) provide the first analytic formulations: for $k$-means, they derive a closed-form move that crosses the Voronoi boundary into a target cluster, and for Gaussian clustering, devise and solve a non-linear formulation. These methods leverage geometry and thus do not apply to density-based algorithms, where membership is defined by density-reachability rather than distances to prototypes. Spagnol et al. (2024) build on BayCon and propose model-specific scores: a distance-based score for $k$-means (scaled distance to the target centroid) and a membership-probability score for HDBSCAN (using the algorithm's soft-clustering outputs). These scores guide a zero-order Bayesian optimisation loop to find CEs. While this makes BayCon-style search applicable to $k$-means and HDBSCAN [1], it provides no density-reachability guarantees and does not explain DBSCAN's core/border/noise status.

Beyond clustering, counterfactuals have been explored for unsupervised outlier detection. Zhou et al. (2025) propose EACE, a method for explaining outliers by converting an outlier into a plausible inlier through minimal modifications. EACE generates one or more counterfactual samples that an anomaly detection model would classify as an inlier, effectively pointing to how an outlying instance could "behave" like the in-distribution data. Similarly, Angiulli et al. (2023) present a contrastive learning approach to repair outliers. Their technique identifies a subset of features and adjusted values (a "mask") that would make an anomalous point no longer isolated, i.e., turn it into an inlier.

Both EACE (Zhou et al., 2025) and the density-contrastive "repair" (Angiulli et al., 2023) methods target outlier detection models that expose a continuous anomaly score (e.g., one-class SVM, kernel density estimators, autoencoders, Isolation Forest/LOF-style scores). Their goal is to decrease the anomaly score so that a point crosses a detector-specific decision threshold and becomes "normal," which is highly relevant for detectors where "inlierness" is defined by that score. DBSCAN does not provide such a score, soft memberships, or gradients; it yields discrete core/border/noise assignments based on $\varepsilon$-reachability and $minPts$. Consequently, these counterfactual outlier detection methods optimise an objective that is orthogonal to ours, as they can make a point "less anomalous"

---

[1] HDBSCAN uses soft memberships (McInnes et al., 2017; Campello et al., 2015) unlike original DBSCAN.

in a detector's sense without guaranteeing that it becomes density-connected to a DBSCAN cluster. In contrast, ExDBSCAN includes both cluster-to-cluster and noise-to-cluster transitions.

# 3 EXPLAINING DBSCAN WITH COUNTERFACTUAL REASONING

Consider data set $X = (x_1, \ldots, x_l), x_i \in \mathbb{R}^n$ and cluster assignment from points $\mathscr{C} : X \to \{-1, 0, \ldots, m-1\}$ to one of $m$ cluster labels, or to noise $(-1)$. The popular density-based clustering approach DBSCAN (Ester et al., 1996) groups points based on a similarity notion modelled as density-reachability. Clusters are maximal areas of density-connected points, separated from other clusters through sparse areas. It uses two parameters: $\varepsilon$, which defines the radius of local density assessment, and $minPts$, which specifies the minimum number of points required to form a dense region. The $\varepsilon$-*neighbourhood* are the points within radius $\varepsilon$ of a point $p$: $\mathscr{N}_\varepsilon(p) = \{q \mid \|p - q\|_2 \leq \varepsilon, \text{ with } p, q \in X\}$. DBSCAN distinguishes between dense *core points*, *border points* close to core points, and *noise points* or outliers, that do not belong to any cluster:

**Definition 3.1** (Core, border, and noise points)**.** The set of *core points* $Q$ are points $p$ that have at least $minPts$ points in their $\varepsilon$-neighbourhood: $Q = \{p \in X \mid |\mathscr{N}_\varepsilon(p)| \geq minPts\}$. Points in $X \setminus Q$ are *border points* $B$, if they have at least one core point in their neighbourhood, else they are *noise*.

**Counterfactual Explanations (CEs).** CEs were first proposed as an optimisation problem by Wachter et al. (2017) for classification models:

**Definition 3.2** (Counterfactual Explanations)**.** A set of $k$ CEs for a point $p$ under a classification model $f$, denoted as $C_k(p)$ minimises a cost function for which the classification outcome changes:

$$C_k(p) = \underset{p'_1, \ldots, p'_k}{\operatorname{argmin}} \, cost(p, p') \text{ subject to } f(p) \neq f(p') \tag{1}$$

Counterfactual generation methods vary in the choice of cost function, typically balancing between desired properties, e.g., the distance between the original point $p$ and the counterfactual $p'$ or the pairwise distance between counterfactuals. In the conemph of density-based clustering, the cost function in Definition 3.2 takes into account the clustering algorithm's concept of similarity.

## 3.1 COUNTERFACTUALS FOR DENSITY-BASED CLUSTERING

Counterfactual explanations provide value for clustering tasks by addressing the interpretability challenges in unsupervised learning. Unlike supervised learning where model decisions can be evaluated against known labels, clustering methods function without predefined classes. This leaves the task of understanding why certain groups are formed or why points are assigned to a particular cluster. CEs bridge this gap by revealing the minimal changes required to alter a point's cluster assignment, thus explaining group differences and influential features. Density-based clustering finds clusters of arbitrary shapes and can identify noise points. These capabilities can make the resulting clusterings difficult to interpret because the complex, non-convex cluster boundaries and the distinction between core, border, and noise points are not immediately obvious. Users cannot easily identify which specific features or density thresholds have led to the assignments. Noise-to-cluster and cluster-to-cluster CEs reveal key features that determine density-based cluster membership.

**Cluster Assignment.** Counterfactual generation requires assigning out-of-dataset points so we can test whether a candidate lies in the target cluster. Because DBSCAN lacks a native assignment for new points, we keep the clustering fixed and define $\mathscr{C}$: point $p$ belongs to cluster $i$ *iff* there exists a core point $q$ with $\mathscr{C}(q) = i$ within the $\varepsilon$-neighbourhood, such that $q \in \mathscr{N}_\varepsilon(p)$.

We restrict membership to core-point neighbourhoods, as placing a CE near a border does not ensure density-connectivity (and may violate DBSCAN's membership definition). Fixing the clustering also avoids the computational and conceptual pitfalls of re-clustering (e.g., merges or new clusters), ensuring we explain the original structure rather than one altered by the CEs.

Generating multiple diverse CEs, rather than a single instance, enhances explanatory value by probing cluster-assignment boundaries from different directions, giving stakeholders a thorough view of the clustering algorithm's behaviour. Multiple examples help surface explanatory trends (e.g.,

features repeatedly changed likely matter for assignment) and reduce perceived arbitrariness, improving trust (Miller, 2019). Regarding actionability (feasibility of the proposed change), options differ by context, e.g., in a medical setting, changing the respiratory rate vs. the oxygen saturation may be more or less realistic for a given patient; thus, offering several paths raises the chance that at least one is realistic and acceptable (Stepin et al., 2021).

It is thus crucial that CEs demonstrate diversity (Laugel et al., 2023). CEs that differ only marginally are redundant and fail to highlight differences among clusters (Mothilal et al., 2020). High redundancy adds no valuable information, for example, two counterfactuals that vary by only $0.1$ breaths per minute in respiratory rate are effectively indistinguishable and convey the same information. While diversity is essential, CEs that stray too far from the explained point risk suggesting unrealistic changes. If a CE bears little in common to the point being explained, human stakeholders may struggle to relate the two and reason about their differences. Proximity therefore remains fundamental to counterfactual reasoning, as it increases both the realism and feasibility while reducing cognitive load for stakeholders, as there are less overall changes (Miller, 2019).

### 3.2 ExDBSCAN: Explaining DBSCAN with counterfactual reasoning

ExDBSCAN addresses these requirements by generating valid sets of counterfactuals that balance proximity and diversity. Users can specify their number and the target cluster to meet their needs.

**Inter-Cluster Concept of Distance.** In CE literature, density is used as a proxy for the ease of moving between points or feasibility (Poyiadzi et al., 2020; Zhou et al., 2025; Yamao et al., 2024; Kanamori et al., 2020). In DBSCAN, cluster members are density-connected, so a counterfactual that moves within a cluster stays in dense regions with similar neighbours, enabling realistic paths. However, because DBSCAN discovers arbitrary shapes, two points can be close in Euclidean[2] yet require a long density-connected path; straight-line distance can therefore overstate similarity.

CEs that look redundant under Euclidean distance, i.e., proposals close in straight-line distance, may still be meaningfully diverse once DBSCAN's density structure is considered. Euclidean close points can lie in different density-reachable regions and thus represent distinct alternatives. Relying on Euclidean distance risks "cutting through" clusters and ignoring connectivity, yielding explanations that misstate feasibility. We illustrate this in Appendix A.6.

To address the challenge and incorporate DBSCAN's notion of similarity, we define proximity as the length of the weighted density-connected path. For each cluster, we build an undirected weighted graph $G(V, E)$ whose vertices $V$ are core points; edges connect two vertices if they are directly density-reachable (distance $< \varepsilon$) and edge weight is their distance. The distance between any two points is then the weighted shortest-path in $G$. Thus, instances are "close" only if an easy path connects them, letting us evaluate diversity while respecting the cluster's internal structure; ignoring this structure can misrepresent separations and, consequently, diversity.

**Reference Core Point.** Given our definition of the assignment function, a CE is assigned to the target cluster iff it falls within the $\varepsilon$-neighbourhood of a cluster's core points. When selecting a point in the space as a CE, we naturally associate it with its nearest core point. To properly leverage the weighted graph $G$ that captures the cluster's structure, we identify the most appropriate reference core point for each proposed CE and generate the CE example within its $\varepsilon$-neighbourhood. Hence, a set of ExDBSCAN CEs correspond directly to a set of vertices in graph $G$, namely the cluster's core points. We denote the set of CEs for point $p$ as $C_k(p)$, while $C'_k(p)$ represents the corresponding set of reference core points.

**Proximity Through Attraction.** By definition, the most proximal CEs are the ones closest to the point to explain $p$. Given $k$ CEs, the set of core points $C'(p)$ maximising proximity will thus be composed of the $k$ nearest core points of $p$. As $p$ and candidate core points are not density connected, the distance metric used is the one employed for fitting DBSCAN.

**Diversity Through Repulsion.** In CEs, a set is diverse when its members are mutually dissimilar (Mothilal et al., 2020). One class of methods enforces this by maximising pairwise distances; variants of the maximum diversity problem (MDP) (Laugel et al., 2023; Ley et al., 2022). An-

---

[2]In this paper, we use the Euclidean distance as it is the most popular metric. A discussion on different metrics is in App. A.4

other models diversity via repulsion, minimising similarity between selected instances; determinantal point processes are common realisations, probabilistically favouring diverse subsets (Mothilal et al., 2020; Kulesza et al., 2012; Afrabandpey & Spranger, 2022).

We observe that simply adopting (Euclidean) distance falls short for DBSCAN: its very notion of similarity is density-connectivity. Thus, the challenge is to model proximity and diversity using density-connectivity. We adopt a repulsion-based approach that prevents redundant counterfactuals, as repulsion forces increase substantially as points converge. In contrast, maximising pairwise distances remains more forgiving toward closely positioned instances, provided that other point pairs maintain sufficient separation to preserve overall diversity. Also, repulsion interactions naturally produce more evenly distributed selections across the solution space. In our density-based setting, we implement this diversity measure using the weighted shortest path distance defined by graph $G$ to evaluate pairwise distances in a cluster. This ensures that our diversity metric respects the cluster's density-connectivity structure rather than relying on simple Euclidean distances that could cut through the cluster and misrepresent the actual relationships between points.

**Balancing Proximity and Diversity.** ExDBSCAN generates multiple CEs that ensure both proximity and diversity; an inherent trade-off since closer alternatives are often less diverse. We resolve this by modelling a physical system and selecting its equilibrium (the minimum-energy configuration), where the proximity-diversity balance is optimal.

For diversity, we treat candidate core points as like-charged particles with Coulomb's electrostatic law (Halliday et al., 2013), pushing already selected cores apart so the minimum-energy set is maximally spread; making the minimum energy configuration the most diverse.

For proximity, we bias chosen core points to be close to the original point, by connecting the latter with a spring to candidate core points. The further the candidate CE, the stronger the force pushing it closer to the original instance, the force scales linearly with the weighted path distance according to Hooke's law (Halliday et al., 2013). Proposition 1 states how the minimum energy configuration caused by the spring-like interactions, selects the closest core points, accounting for proximity.

The aim is to find the system's stable configuration, i.e., the one that minimises the energy of the physical system. Spring-like forces bias the system to select points closest to the original instance as springs push towards their rest state; i.e., when explained point and candidate core point have a null distance. Electrostatic-like repulsion favours core points to be pairwise distant (according to the shortest weighted path distance), as same-charged particles push each other away. The minimum energy configuration balances the springs' desire to go back to their rest configuration with the charges' desire to spread as much as possible. Hence, given a set $C'(p)$ of chosen core points, and point to explain $p$, the energy configuration of the system has the form:

$$E_{C'} = \sum_{V_i \in C'} \sum_{V_j \in C':j>i} \frac{1}{\mathscr{D}(V_i, V_j)} + \sum_{V_i \in C'} d^2(p, V_i) \tag{2}$$

$V_i$ is the $i$th vertex of graph $G$, $\mathscr{D}(V_i, V_j)$ denotes the weighted shortest path distance between vertices $V_i$ and $V_j$. Lastly, $d(p, V_i)$ is the Euclidean distance between the point to explain $p$ and vertex $V_i$.[3] We evaluate physical constants characterising spring and electrostatic terms in Eq. 2 as 1, as we are only interested in how energy terms scale with distance. To find the optimal configuration of the system and thus, the set of $\mathscr{C}$, we find the subset $\mathscr{C}'$ of vertices $V$ minimising system energy:

$$C'_k(p) = \operatorname*{argmin}_{B \in V : |B|=k} E_B \quad . \tag{3}$$

The energy term in Eq. 2 can be seen as ExDBSCAN's cost function in the formalism introduced in Eq. 1. Fig. 1 shows how a solution of Eq. 3 looks like on a toy dataset where counterfactuals are produced for a noise point. On the left of Fig. 1, considering the electrostatic term only leads to maximally diverse counterfactuals. On the right, using just spring-like terms make counterfactuals maximally proximal. In the middle, balancing repulsion and attraction considering the cluster's density structure, prompts proximal and spread-out counterfactuals. We observe the following:

**Proposition 1.** *Aligning with our desiderata on prixmity, if considering only the spring-like term in Eq. 2, solving the optimisation problem in Eq. 3 is equivalent to selecting the $k$ nearest neighbours of the point to explain $p$.*

---

[3]Depending on the dataset's and cluster's structure distances $d(p, V_i)$ and $\mathscr{D}(V_i, V_j)$ might have different scales. Because of this, we use a normalisation scheme introduced in Appendix A.3.

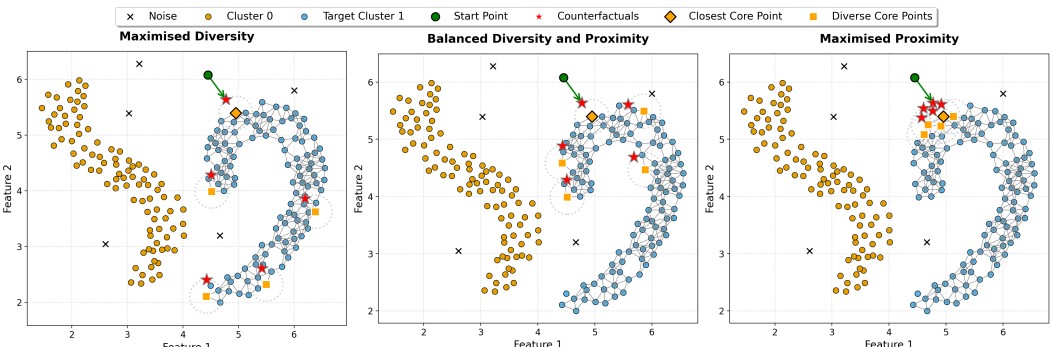

Figure 1: Optimising diversity (left), balanced (middle), optimising proximity (right). Cluster points in yellow and blue, resp.; noise marked with X; green point to be explained; red stars counterfactuals.

*Proof.* From Eq. 2, and by induction, when the set of core points is empty, the core minimising the elastic energy of the system is the closest to the point to explain $p$ (minimum possible distance). As far as the induction step, given a set of $k' < k$ selected core points, the next core point to be selected, i.e., the one minimising the total elastic energy, is given by the closest non-selected core-point $p$. ☐

Solving the optimisation problem in Eq. 3 is computationally challenging.

**Proposition 2.** *Solving Eq. 3, i.e. the energy minimisation problem with both attraction and repulsion, is an NP-hard problem.*

*Proof.* (Sketch, full in Appendix A.7) Maximum diversity problems are NP-hard (Parreño et al., 2021). Eq. 3 restricted to the electrostatic term can be mapped to the MDP. Adding springs does not change NP-hardness: If it did, the MDP could be solved in polynomial time sending spring constants to zero. ☐

We approximate the solution using the following greedy algorithm. Given an incomplete set of selected core points $C'_k$ with $k' < k$, the next cluster point is chosen as the one minimising the total energy of the system according to Eq. 2. When choosing the instance initialising the set of core points, $C'$ is still empty and thus no repulsive electrostatic-like force influences the choice. $C'$ will thus be initialised with the core point closest to the point to explain since it is the one greedily minimising the energy configuration, composed by a single spring-like term. Note in the case with only one counterfactual, the greedy optimisation corresponds to the optimal one according to Eq. 2.

**Choosing the set of counterfactuals** $C$**.** Our chosen set of core points $C'$ leverages the clusters' density structure to optimally balancing proximity and diversity through the weighted undirected graph $G$. Given $C'$, we need to select where in each core point's $\varepsilon$ neighbourhood the counterfactual is going to be placed, practically mapping $C'$ to the final set of counterfactuals $C$. Given a single core point $q$ in $C'$, to further improve proximity and respecting our assignment function definition, we position the counterfactual $p' \in C$, $\varepsilon$ away in the direction of the original point $p$:

$$p' = p + (d_{pq} - \varepsilon)(q - p)/d_{pq}.e \tag{4}$$

The counterfactual point $p'$ is considered part of the cluster containing reference core point $q$, as it satisfies the density-based clustering criteria for the target cluster, effectively ensuring validity.

**Theorem 3.1.** *Every counterfactual point $p'$ with reference core point $q$, such that $\mathscr{C}(q) = i$, $i \neq -1$, that is generated for a point $p \in X : \mathscr{C}(p) \neq i$ toward cluster $i$ with the proposed method, is considered part of cluster $i$, implying that $\mathscr{C}(p') \neq \mathscr{C}(p)$ and $C(p') = i$, and thus valid counterfactual.*

*Proof.* (Sketch) The definition in Section 3.1 denotes that $p'$ is in the same cluster as its reference core point $q$ with $\mathscr{C}(q) = i$, therefore $\mathscr{C}(p') \neq \mathscr{C}(p)$ and specifically $\mathscr{C}(p') = i$. ∎

How to retrieve the set of counterfactuals $C$ when a set of features is non-actionable is detailed in Appendix A.5 where we show how ExDBSCAN creates a graph only considering reference core points whose $\varepsilon$ neighbourhood is reachable without changing non-actionable features.

## 4 EXPERIMENTS

**Experimental Setup.** We study 30 OpenML tabular datasets (Vanschoren et al., 2013)[4] to assess counterfactuals in terms of being *valid*, *proximal*, and *diverse*. For each dataset we find the best $(\varepsilon, \text{minPts})$ by grid search wrt. DBCV score (Moulavi et al., 2014). Details in App. in Table 1. We compare ExDBSCAN to BAYCON, a model-agnostic counterfactual method. BAYCON treats the clustering as a black-box prediction task, repeatedly querying it with candidate points and using Bayesian optimisation to search the feature space. BAYCON finds counterfactuals, but does not utilise DBSCAN's density-connectivity. BAYCON is a state-of-the-art model-agnostic competitor, but lacks the guarantees and structural awareness that ExDBSCAN provides for obtaining multiple CEs. For each cluster $C_i$ or noise partition $N$, sample 10 points randomly. For each sampled point $p \in C_i$, create a target for every other cluster $C_j$ with $j \neq i$ yielding $10 \times \sum_{i=1}^{m}(m-1) + 10 \times m$ queries per dataset. Each query asks to produce a set of $k = 10$ CEs in the specified target cluster.

1) **Validity ($\uparrow$):** Proportion of returned $k$ counterfactuals that reach the target cluster (i.e., the CE is part of the target cluster). When no CE is found, validity is zero for that query.

2) **Average Proximity ($\downarrow$):** Mean distance from original point $p$ to each valid returned CE, measured in DBSCANs clustering feature space. Lower values indicate that the CE more closely resembles the explained point.

3) **Average Diversity ($\uparrow$).** We want $k$ CEs to offer diverse solutions in the target cluster, not minor variants of the same information. We therefore measure the average similarity (defined as the inverse of the pairwise weighted shortest path distance), and use its inverse to report diversity.

For a compact, per-dataset comparison we visualise three stacked barbell plots (Fig. 2): *Validity* (top, higher is better), *Average Proximity* (middle, lower is better), and *Average Diversity* (bottom, higher is better). Datasets are ordered by BAYCON validity (descending). For each dataset, connectors link *BayCon* (red markers) and ExDBSCAN (blue markers) to emphasise within-dataset differences.

**Validity Results.** In the top view in Figure 2, ExDBSCAN obtains perfect validity on all datasets, meaning every generated CE successfully attains the target cluster assignment. In contrast, BAYCON shows substantial variability in validity, with scores ranging from near-zero on several datasets to $\approx 0.8$ on the best-performing datasets. On the *census6* dataset, BAYCON even entirely fails to produce any valid counterfactual. These differences reflect the fundamental distinction in how the methods operate. ExDBSCAN's validity guarantee stems from its construction by following density-reachable paths through the graph and positioning CEs within $\varepsilon$-neighbourhoods of target core points. The experiments thus empirically confirm the validity by design as stated in Theorem 3.1. Every step of the CE generation process respects DBSCAN's density-connectivity structure, guaranteeing that the final CE satisfies the cluster membership criteria.

BAYCON, conversely, treats the clustering as a black-box prediction task and explores the feature space using Euclidean distance without awareness of DBSCAN's connectivity requirements. It can propose changes that move points toward target clusters in a straight-line distance, but do not achieve the density-connectivity that DBSCAN requires for membership. This is precisely why a DBSCAN-specific method proves valuable, because model-agnostic approaches lack the structural knowledge to reliably generate valid density-based CEs. The inferior results from BAYCON are thus not due to poor optimisation, but rather due to a fundamental mismatch between the Euclidean search strategy and DBSCAN's density-based decision boundaries.

**Proximity Results.** The middle view in Figure 2, reveals that ExDBSCAN consistently produces more proximal counterfactuals than BAYCON across all datasets. When comparing only valid CEs from both methods, ExDBSCAN achieves lower proximity scores in every case. The superiority in proximity is notable given ExDBSCAN's perfect validity as this means that our method does not

---

[4]Dataset details and reproducible code: Link to repository.

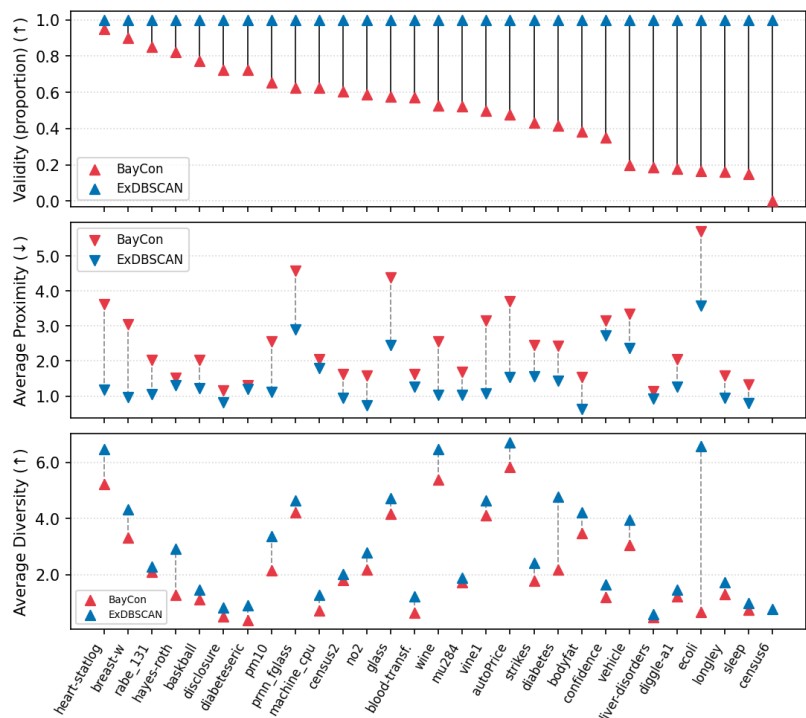

Figure 2: **CE quality metrics for** 30 **datasets (x-axis).** BAYCON (red) and EXDBSCAN (blue) barbells on same dataset. *Top:* Validity (proportion of counterfactuals that attain target; ▲ better). *Middle:* Avg. proximity of valid CEs (▼ better). *Bottom:* Avg. diversity as mean distance (▲ better).

sacrifice one for the other. The physics-inspired optimisation balances the spring-like forces pulling CEs toward the original point with the repulsion-like forces ensuring diversity. BAYCON'S higher proximity values suggest it often makes larger less targeted changes to achieve cluster reassignment, likely because it cannot leverage the density-connected pathways that EXDBSCAN does.

**Diversity Results.** The bottom view in Figure 2 shows that EXDBSCAN generates more diverse sets than BAYCON, as measured by mean pairwise distance within the density-reachability graph. This diversity advantage is meaningful because it indicates that EXDBSCAN'S CEs span different areas of the target cluster. The diversity arises from the repulsion component of the energy system, which pushes selected core points apart according to their graph-based distance. BAYCON'S lower diversity scores suggest its CEs tend to concentrate in limited areas of the feature space, providing more redundancy and thus less comprehensive exploration of alternative ways to achieve cluster membership. Note, EXDBSCAN achieves superior diversity, better proximity and perfect validity.

Due to lack of space, we defer additional experiments that demonstrate that EXDBSCAN also performs very well when a set of features is non-actionable and thus not helpful in CEs, to App. A.9.2.

## 5 CONCLUSION AND FUTURE WORK

We proposed ExDBSCAN, a novel counterfactual reasoning framework for density-based clustering. In extensive experiments, we showcase that ExDBSCAN generates diverse, valid, and proximal counterfactuals. With our physics-inspired diversity model we balance diversity and proximity when generating sets of counterfactuals. ExDBSCAN enhances the application of DBSCAN to ensure interpretability and thus user trust. Future work concentrates on generating counterfactuals for incremental density-based clustering as necessary for, e.g., streaming data.

## REPRODUCIBILITY STATEMENT

We include an anonymised GitHub, including all code necessary to reproduce our experiments.

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

# A APPENDIX

## A.1 LLM STATEMENT

No generative AI tools were used in the writing of the paper and at any stage of this research.

## A.2 ALGORITHM PSEUDOCODE

To complement Section 3.2 we include full pseudocode for the following two components of ExDB-SCAN:

1. Selection of $k$ reference core points via greedy minimisation of energy function in Eq. 3.

2. Generation of counterfactual examples from the selected core points using Eq. 4

These algorithms make explicit the operations described in the main text and show how proximity and diversity are jointly optimised.

### A.2.1 GREEDY SELECTION OF CORE POINTS

Given a point $p$ to be explained and a target cluster, the first step of ExDBSCAN is to select a set of $k$ core points whose $\varepsilon$-neighbourhoods will serve as the basis for constructing counterfactuals. The energy of a candidate set $C'$ is:

$$E_{C'} = \sum_{V_i \in C'} \sum_{V_j \in C':j>i} \frac{1}{\mathscr{D}(V_i, V_j)} + \sum_{V_i \in C'} d^2(p, V_i).$$

The optimisation problem in Eq. 3 is NP-hard (Proposition 2), so we use a greedy approximation. For $k = 1$, this procedure returns the optimal solution.

---

**Algorithm 1** Greedy Selection of Core Points

---

**Require:** Point to explain $p$; set of core points $V$; number of counterfactuals $k$; weighted graph distance $\mathscr{D}(\cdot, \cdot)$; feature-space distance $d(\cdot, \cdot)$; target cluster $t$.
**Ensure:** Set $C'$ of $k$ selected core points
1: $C' \leftarrow \emptyset$
2: $V^t = \{V_j : V_j \in t\}$
3: **while** $|C'| < k$ **do**
4:     best_core $\leftarrow$ None,   best_energy $\leftarrow +\infty$
5:     **for** each core point $V_j \in V \setminus C'$ **do**
6:         $S \leftarrow C' \cup \{V_j\}$
7:         $E_S \leftarrow 0$                                     ▷ Repulsion term (graph distances)
8:         **for all** pairs $(V_i, V_\ell)$ in $S$ with $\ell > i$ **do**
9:             $E_S \leftarrow E_S + \dfrac{1}{\mathscr{D}(V_i, V_\ell)}$
10:        **end for**                              ▷ Attraction term (feature-space distance)
11:        **for all** $V_i \in S$ **do**
12:           $E_S \leftarrow E_S + d^2(p, V_i)$
13:        **end for**
14:        **if** $E_S <$ best_energy **then**
15:           best_energy $\leftarrow E_S$
16:           best_core $\leftarrow V_j$
17:        **end if**
18:     **end for**
19:     $C' \leftarrow C' \cup \{\text{best\_core}\}$
20: **end while**
21: **return** $C'$

---

All shortest-path distances $\mathscr{D}(V_i, V_j)$ are precomputed once for the target cluster. Evaluating each candidate core point requires $O(|C'|)$, yielding a total complexity of $O(km)$ where $m = |V|$.

### A.2.2 Constructing Counterfactuals

Once the set of reference core points $C' = \{q_1, \ldots, q_k\}$ has been selected, each counterfactual $p'$ is placed inside the $\varepsilon$-neighbourhood of its corresponding core point $q$. To maximise proximity to the original point $p$, we move away from $p$ towards $q$ and stop at distance $\varepsilon$ from $q$. This corresponds exactly to Eq. 4 in the main text. Because each constructed point lies within $\varepsilon$ of a core point in the target cluster, Theorem 3.1 guarantees that all counterfactuals are valid. Algorithm 2 summarises the procedure through pseudocode.

---

**Algorithm 2** Generate Counterfactuals from Selected Core Points

---

**Require:** Point $p$; selected core points $C' = \{q_1, \ldots, q_k\}$; DBSCAN radius $\varepsilon$; feature-space metric $d(\cdot, \cdot)$

**Ensure:** Set $C$ of $k$ counterfactuals
1: $C \leftarrow \emptyset$
2: **for all** $q_j \in C'$ **do**
3:      $d_{pq} \leftarrow d(p, q_j)$         ▷ By construction, $d_{pq} > 0$ since $p$ and $q_j$ belong to different clusters
4:      direction $\leftarrow (q_j - p)/d_{pq}$
5:      $p' \leftarrow p + (d_{pq} - \varepsilon) \cdot$ direction                            ▷ Eq. 4
6:      $C \leftarrow C \cup \{p'\}$
7: **end for**
8: **return** $C$

---

Because each $p'$ lies within distance $\varepsilon$ of a core point in the target cluster, the construction ensures all counterfactuals are valid.

### A.2.3 Target Cluster Not Specified

Pseudocode 1 requires the target cluster in input. The user could prefer not to specify a certain target cluster, e.g. if the user wants to convert a noise point to a generic non-noise instance. In this case, pseudocode 1 would need to be slightly changed on line 8. Specifically, instead of adding a repulsion term for each pair of core points already selected unconditionally, we add it only if they belong to the same cluster. Points belonging to different cluster are indeed already diverse by construction and they are not connected through any density connected path according to DBSCAN's algorithm.

### A.3 Normalisation Scheme

The distance $d(p, V_i)$ from candidate core points $V_i$ to the explained point $p$ and the distances between candidate core points $\mathscr{D}(V_i, V_j)$, could have different scales depending on the the dataset, with $d(p, V_i)$ potentially being bigger as it connects points that are not in a cluster. In order to prevent the spring-like term to dominate, given the distance $d_c$ from $p$ to the closest core point, and the average graph weight $\bar{\mathscr{D}}$, we scale distances $\mathscr{D}(V_i, V_j)$ by $d_c/\bar{\mathscr{D}}$.

### A.4 Metric

By design, ExDBSCAN is agnostic to metrics. That being said, if DBSCAN is fit on a metric different to euclidean, e.g., Minkowski, cosine, Mahalanobis, or mixed-type metrics, ExDBSCAN can be easily adjusted to this. Instead of using euclidean distance, we can employ any other distance metric to determine reference core-points as well as solving the energy-minimisation to produce diverse and proximal counterfactuals. We also highly advise, to then also use the using the same metric in ExDBSCAN that was selected for DBSCAN clustering to maintain consistency in the concepts of density-reachability and graph connectivity. Furthermore, we advise employing this chosen metric to evaluate proximity and diversity, as the underlying structure of the data and clustering depends on it.

### A.5 NON-ACTIONABLE FEATURES

Actionability ensures that the generated CEs are actionable, i.e., a user can actually realistically change the features. Non-actionable features as genetic information or inherent personal attributes, including age and gender, make explanations unrealistic and prevent human stake-holders to simulate and apply the proposed changes.

#### A.5.1 NON-ACTIONABLE FEATURES IN EXDBSCAN

To account for non-actionable features, we restrict the set of core points to consider, when identifying the set $C'$. The constrained set of core points includes all core points that are not further than $\varepsilon$ away from the original point when measuring distances in the subspace defined by the non-actionable features $\mathbb{R}^{n'}$, with $n' < n$ the number of non-actionable features. In this way, we take into consideration only core points whose $\varepsilon$ neighbourhood is reachable without changing any of the non-actionable features.

The undirected weighted graph $G$ is then built and the set of reference core points chosen $C'$ as described in Section 3.2. When moving from the set of core points $C'$ to the set of counterfactuals $C$, it is not guaranteed that we can move in the direction of the point to explain $p$, as we now have the constraint given by the non-actionable features. Therefore, we move toward point $p$ only in the subspace $\mathbb{R}^{n-n'}$ formed by the actionable features, still putting the counterfactual $\varepsilon$ away from the reference core point, which is always possible given our core point filtering procedure. We visualise the approach in Figure 3 where we show how a single counterfactual is produced for a toy dataset when feature $Y$ is non-actionable.

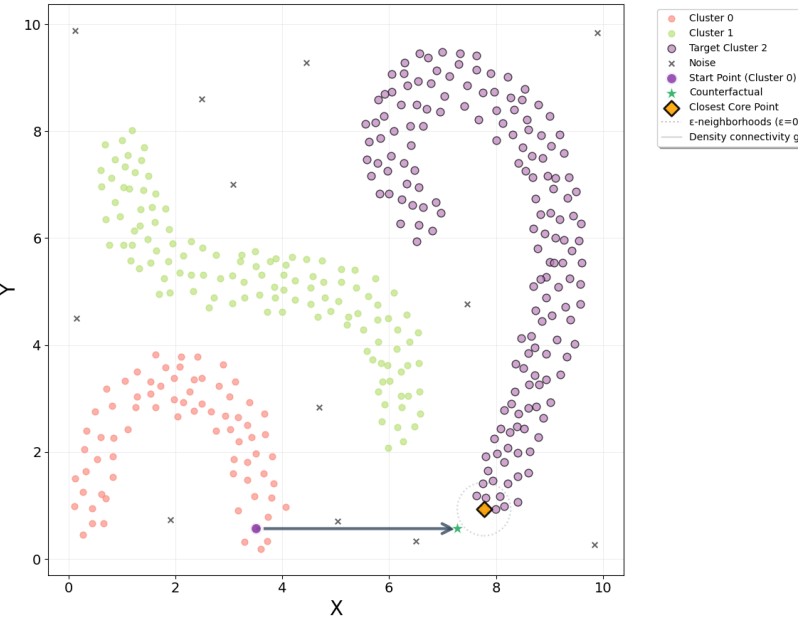

Figure 3: Conceptual Figure illustrating actionability. Coloured points denote cluster points assigned to different clusters. The x-axis shows the actionable and y-axis shows non-actionable feature. The orange rhombus shows the closest core point, that is in the non-actionable direction not further than $\varepsilon$ away from the point to explain (purple). The green star is the identified counterfactual.

#### A.5.2 REAL-WORLD CONSTRAINTS

There are different strategies to account for real-world constraints. First of all, if the non-actionable features include categorical features, we suggest changing the metric employed in DBSCAN as well as ExDBSCAN, for more details to employing different metrics see Section A.4.

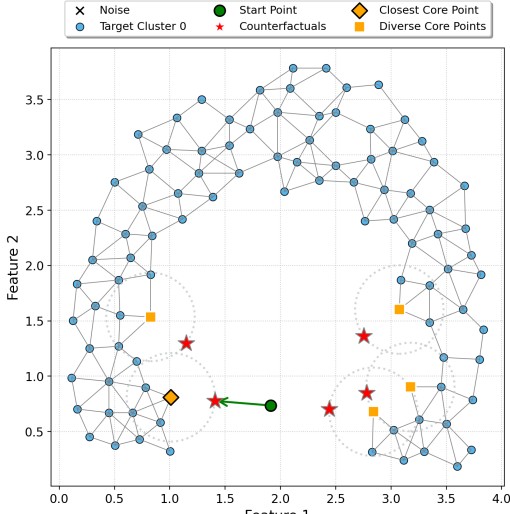

Figure 4: Counterfactuals (red stars) with their closest core point (orange squares) for a noise point (green) toward a half circle cluster. Counterfactuals near the opposite ends of the half circle are Euclidean near, moving between the two arcs ends would instead require many intermediate steps, as they are far in terms of DBSCAN's density-connected path

Other constraints, i.e., monotonicity or feature bounds can be easily incorporated through further filtering of the core point set. Instead of restricting the set of reference core points to actionable dimensions only, we can additionally restrict this to be a specific direction or range depending on the constraints defined by the user. This allows us to also incorporate already known domain knowledge into our model.

## A.6 EUCLIDEAN DISTANCE'S LIMITATIONS

Instances that appear redundant based on Euclidean distance, might add valuable diversity when considering the cluster's structure. Using our shortest path weighted distance properly takes into account the clusters' structure. We exemplify this in Figure 4, where counterfactuals are proposed for a noise point toward a half-circle cluster. Points near the opposite ends of the arc appear close in the Euclidean space, but under DBSCAN they are density distant; a critical distinction when evaluating diversity for CEs.

## A.7 NP-HARDNESS

Proposition 2, states that the optimisation problem in Equation 3 is *NP-hard*, we present here a more extended proof.

*Proof.* Consider the energy landscape in Equation 2, the electrostatic term reads (in contracted notation) $\sum_{i,j>i} \frac{1}{\mathscr{D}(V_i, V_j)}$ , which needs to be minimised to find the optimal energy configuration. Minimising the sum of the energy terms is equivalent to maximise their negative. Thus, the problem can be mapped to a maximum diversity problem where the object summed is the inverse distance, MDP optimisation is known to be NP-hard (Parreño et al., 2021). Consider now adding the spring terms, considering Equation 2 in full. We can again map the minimisation problem to a maximisation one by taking the negative energy. If this new optimisation problem would be solvable in polynomial time, we could move the spring constants (which we consider equal to 1) to 0. In this limit, the MDP could be also solved in polynomial time, contradicting its NP-hardness. ∎

## A.8 EXPERIMENTAL SETUP

### A.8.1 SURROGATE MODELS

While training surrogate classifiers on DBSCAN labels might seem like the straightforward baseline, we decided to not include this due to the following reasons:

- **Noise handling**: DBSCAN does not only group the points into clusters, but it additionally defines points that do not belong to a cluster as noise points. The assigned noise points do not form a coherent "class" with meaningful decision boundaries, they maybe scattered around the complete dataset. Consequently, any surrogate classifier would need to learn from highly irregular or disconnected regions, which would likely result in poor fidelity.

- **Surrogate complexity**: Constructing a surrogate introduces drawbacks and uncertainty on its own, additional hyperparameters, tuning steps, and optimisation challenges. As a consequence, the comparison between the models is less controlled and less interpretable.

- **Lack of fidelity guarantees**: Even if the surrogate would have a high accuracy, there is no guarantee that it reproduces DBSCAN's density-reachability criteria reliably. Incorrectly learned boundaries would lead to counterfactuals that are "valid" for the surrogate but not for DBSCAN. While the CEs would explain the classifiers / surrogates decision boundaries, this is not directly transferable and therefore also not applicable as an evaluation strategy.

- **Instability concerns**: Prior work Visani et al. (2022) shows that surrogate models trained on synthetic or held-out data can be unstable, especially in settings with complex or non-convex decision boundaries such as density-based clustering.

We see that comparing to a surrogate model poses an interesting question, regarding the differences in information extracted between the surrogate model and DBSCAN itself. While we do not see this as an evaluation framework, we plan future work employing ExDBSCAN as a tool to investigate those differences.

## A.9 EXPERIMENTAL RESULTS

The following sections present the complete numerical results in tables, thereby we distinguish between results regarding actionability (Sect. A.9.2) and results without non-actionable features (Sect. A.9.1).

### A.9.1 ALL FEATURES ARE ACTIONABLE

**Validity**    Given a point to explain, a method produces a valid outcome if it returns a counterfactual, and the counterfactual actually belongs to the cluster. High validity is thus a proxy of how a method effectively deals with challenging scenarios where a counterfactual is difficult to retrieve.

To evaluate validity we take the results used to assess proximity in the following paragraph and compute the proportion of valid counterfactuals. Results are displayed in Table 1.

**Proximity**    Table 1 shows how EXDBSCAN consistently outperforms BAYCON in terms of proximity. Leveraging DBSCAN's structure, EXDBSCAN is able to find the best counterfactual across all datasets. The more proximal the counterfactual, the more the counterfactual probes the local decision boundary of the model, increasing the probability the the explanation reflects the real reason why the model made its decision.

**Diversity**    Table 1 showcases how EXDBSCAN outperforms BAYCON regarding diversity.

### A.9.2 NON-ACTIONABLE FEATURES

We here evaluate EXDBSCAN in the setting where a subset of features are non-actionable. We utilise the experimental pipeline used in Section 4, additionally choosing a random set of features as non actionable. How many features are non actionable is a parameter chosen at random between one 1 and $\frac{n}{2}$ features, with $n$ the total number of features. Figure 5 and Table 2 showcase results.

Table 1: Complete results regarding *Proximity*, *Validity*, and *Diversity* for ExDBSCAN (ours) and BayCon. Accross all experiments ExDBSCAN achieved better (higher) validity and diversity, and better (lower) proximity. Mean and standard error of the mean are reported. The standard error of the mean is not reported for validity as it is a metric characterising the whole set of counterfactuals and not each counterfactual. We write n.a. when there are not enough samples to estimate the standard error of the mean.

| Dataset | Method | Proximity | Validity | Diversity |
|---|---|---|---|---|
| autoPrice | ExDBSCAN | $1.5 \pm 0.2$ | 1.00 | $6.71 \pm 0.003$ |
| autoPrice | BayCon | $3.7 \pm 0.2$ | 0.48 | $5.85 \pm 0.007$ |
| baskball | ExDBSCAN | $1.2 \pm 0.1$ | 1.00 | $1.47 \pm 0.02$ |
| baskball | BayCon | $2.0 \pm 0.1$ | 0.78 | $1.13 \pm 0.03$ |
| blood-transfusion-service-center | ExDBSCAN | $1.3 \pm 0.2$ | 1.00 | $1.23 \pm 0.04$ |
| blood-transfusion-service-center | BayCon | $1.6 \pm 0.3$ | 0.57 | $0.66 \pm 0.1$ |
| bodyfat | ExDBSCAN | $0.6 \pm 0.4$ | 1.00 | $4.23 \pm 0.009$ |
| bodyfat | BayCon | $1.5 \pm 0.3$ | 0.38 | $3.47 \pm 0.01$ |
| breast-w | ExDBSCAN | $1.0 \pm 0.1$ | 1.00 | $4.32 \pm 0.01$ |
| breast-w | BayCon | $3.0 \pm 0.2$ | 0.90 | $3.31 \pm 0.01$ |
| chscase_census2 | ExDBSCAN | $1.0 \pm 0.1$ | 1.00 | $2.02 \pm 0.01$ |
| chscase_census2 | BayCon | $1.6 \pm 0.1$ | 0.61 | $1.81 \pm 0.01$ |
| chscase_census6 | ExDBSCAN | $1.1 \pm 0.1$ | 1.00 | $0.78 \pm 0.04$ |
| chscase_census6 | BayCon | - | 0.00 | - |
| chscase_vine1 | ExDBSCAN | $1.1 \pm 0.2$ | 1.00 | $4.642 \pm 0.008$ |
| chscase_vine1 | BayCon | $3.2 \pm 0.3$ | 0.50 | $4.114 \pm 0.009$ |
| confidence | ExDBSCAN | $2.73 \pm 0.05$ | 1.00 | $1.64 \pm 0.01$ |
| confidence | BayCon | $3.15 \pm 0.05$ | 0.35 | $1.20 \pm 0.07$ |
| diabetes | ExDBSCAN | $1.4 \pm 0.2$ | 1.00 | $4.792 \pm 0.005$ |
| diabetes | BayCon | $2.4 \pm 0.1$ | 0.42 | $2.17 \pm 0.04$ |
| diabetes_numeric | ExDBSCAN | $1.2 \pm 0.1$ | 1.00 | $0.90 \pm 0.08$ |
| diabetes_numeric | BayCon | $1.3 \pm 0.1$ | 0.72 | $0.38 \pm 0.2$ |
| diggle_table_a1 | ExDBSCAN | $1.3 \pm 0.1$ | 1.00 | $1.46 \pm 0.02$ |
| diggle_table_a1 | BayCon | $2.1 \pm 0.4$ | 0.18 | $1.22 \pm 0.03$ |
| disclosure_x_noise | ExDBSCAN | $0.8 \pm 0.1$ | 1.00 | $0.82 \pm 0.08$ |
| disclosure_x_noise | BayCon | $1.2 \pm 0.1$ | 0.72 | $0.50 \pm 0.2$ |
| ecoli | ExDBSCAN | $3.6 \pm 0.1$ | 1.00 | $6.59 \pm 0.003$ |
| ecoli | BayCon | $5.7 \pm n.a.$ | 0.17 | $0.67 \pm 0.5$ |
| glass | ExDBSCAN | $2.5 \pm 0.3$ | 1.00 | $4.734 \pm 0.008$ |
| glass | BayCon | $4.4 \pm 0.5$ | 0.57 | $4.166 \pm 0.008$ |
| hayes-roth | ExDBSCAN | $1.30 \pm 0.05$ | 1.00 | $2.92 \pm 0.01$ |
| hayes-roth | BayCon | $1.53 \pm 0.09$ | 0.82 | $1.28 \pm 0.06$ |
| heart-statlog | ExDBSCAN | $1.2 \pm 0.1$ | 1.00 | $6.488 \pm 0.009$ |
| heart-statlog | BayCon | $3.6 \pm 0.2$ | 0.95 | $5.224 \pm 0.004$ |
| iris | ExDBSCAN | $1.2 \pm 0.2$ | 1.0 | $2.774 \pm 0.005$ |
| iris | BayCon | $2.7 \pm 0.2$ | 0.75 | $1.82 \pm 0.03$ |
| liver-disorders | ExDBSCAN | $0.9 \pm 0.3$ | 1.00 | $0.59 \pm 0.07$ |
| liver-disorders | BayCon | $1.1 \pm 0.2$ | 0.19 | $0.49 \pm 0.07$ |
| longley | ExDBSCAN | $1.0 \pm 0.2$ | 1.00 | $1.74 \pm 0.04$ |
| longley | BayCon | $1.6 \pm 0.2$ | 0.16 | $1.31 \pm 0.05$ |
| machine_cpu | ExDBSCAN | $1.8 \pm 0.3$ | 1.00 | $1.27 \pm 0.05$ |
| machine_cpu | BayCon | $2.0 \pm 0.2$ | 0.62 | $0.73 \pm 0.1$ |
| mu284 | ExDBSCAN | $1.0 \pm 0.1$ | 1.00 | $1.89 \pm 0.2$ |
| mu284 | BayCon | $1.7 \pm 0.1$ | 0.52 | $1.74 \pm 0.2$ |
| no2 | ExDBSCAN | $0.7 \pm 0.2$ | 1.00 | $2.78 \pm 0.03$ |
| no2 | BayCon | $1.6 \pm 0.2$ | 0.59 | $2.17 \pm 0.04$ |
| pm10 | ExDBSCAN | $1.1 \pm 0.1$ | 1.00 | $3.369 \pm 0.009$ |
| pm10 | BayCon | $2.6 \pm 0.2$ | 0.66 | $2.15 \pm 0.1$ |
| prnn_fglass | ExDBSCAN | $2.9 \pm 0.4$ | 1.00 | $4.647 \pm 0.009$ |
| prnn_fglass | BayCon | $4.6 \pm 0.6$ | 0.62 | $4.238 \pm 0.007$ |
| rabe_131 | ExDBSCAN | $1.1 \pm 0.1$ | 1.00 | $2.29 \pm 0.01$ |
| rabe_131 | BayCon | $2.0 \pm 0.1$ | 0.85 | $2.11 \pm 0.01$ |
| sleep | ExDBSCAN | $0.8 \pm 0.2$ | 1.00 | $0.98 \pm 0.04$ |
| sleep | BayCon | $1.34 \pm 0.09$ | 0.15 | $0.76 \pm 0.03$ |
| strikes | ExDBSCAN | $1.57 \pm 0.03$ | 1.00 | $2.431 \pm 0.003$ |
| strikes | BayCon | $2.45 \pm 0.04$ | 0.43 | $1.77 \pm 0.006$ |
| vehicle | ExDBSCAN | $2.4 \pm 0.7$ | 1.00 | $3.95 \pm 0.01$ |
| vehicle | BayCon | $3 \pm 2$ | 0.20 | $3.05 \pm 0.04$ |
| wine | ExDBSCAN | $1.0 \pm 0.2$ | 1.00 | $6.476 \pm 0.009$ |
| wine | BayCon | $2.6 \pm 0.4$ | 0.53 | $5.397 \pm 0.008$ |

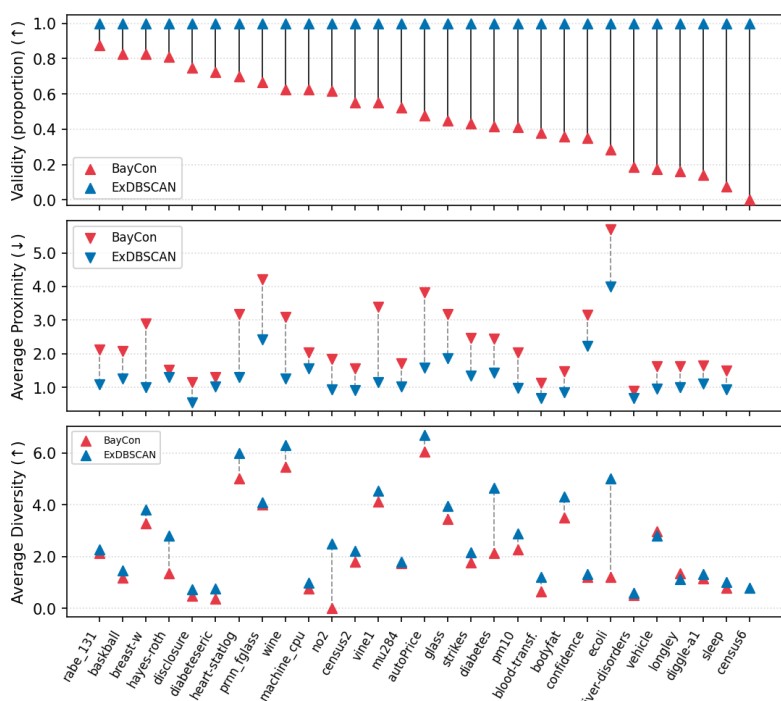

Figure 5: **CE quality metrics for** 30 **datasets (x-axis).** BAYCON (red) and EXDBSCAN (blue) barbells on same dataset. *Top:* Validity (proportion of counterfactuals that attain target; ▲ better). *Middle:* Avg. proximity of valid CEs (▼ better). *Bottom:* Avg. diversity as mean distance (▲ better).

ExDBSCAN demonstrates perfect validity of 1.00 across all experiments, indicating that our approach always generates a CE whenever a valid counterfactual exists. This is not always guaranteed, as there may be instances, where no actionable feature can be changed to alter the cluster assignment. Furthermore, we observe that the produced counterfactuals are generally more proximal than the ones identified with BAYCON. Finally, the lower section of Figure 5, as well as the most right column in Table 2 provide details regarding diversity. ExDBSCAN performs better, with the sole exceptions being the vehicle and longley datasets, where the differences are minimal. In these cases ExDBSCAN still achieves superior results for proximity and validity.

## A.10 RUN-TIME ANALYSIS

To show ExDBSCAN's superior run-time performance over BayCon we run both methods and measure the average time in seconds per-dataset needed to find counterfactuals. Table 3 testifies how ExDBSCAN is significantly faster than BayCon on the tested datasets. In fairness to BayCon, the Bayesian method produces more than the 10 counterfactuals considered but the inability to restrict the counterfactual search is a feature of Baycon.

## B SURROGATES

As an additional benchmark we take into consideration the following scenario. A surrogate model is fitted to the DBSCAN clustering assignment using DBSCAN's cluster labels. Then, a state-of-the-art counterfactual generator, i.e. DiCE is tasked to produce 10 counterfactuals. The surrogate chosen is a feed-forward neural network: a differentiable model allowing gradient-based optimisations. Results displayed in Table 4 show how, compared with ExDBSCAN's, DiCE's counterfactuals are on average significantly more distant to the original instance as showed by the proximity values. DiCE's worse proximity is not balance by a better diversity than ExDBSCAN, with the two methods performing similarly across datasets.

Table 2: Complete results with non-actionable features regarding *Proximity*, *Validity*, and *Diversity* for ExDBSCAN (ours) and BayCon. Accross all experiments ExDBSCAN achieved better (higher) validity and diversity, and better (lower) proximity. Mean and standard error of the mean are reported. The standard error of the mean is not reported for validity as it is a metric characterising the whole set of counterfactuals and not each counterfactual. We write n.a. when there are not enough samples to estimate the standard error of the mean.

| Dataset | Method | Proximity | Validity | Diversity |
|---|---|---|---|---|
| autoPrice | ExDBSCAN | $1.6 \pm 0.2$ | 1.00 | $6.701 \pm 0.003$ |
| autoPrice | BayCon | $3.8 \pm 0.2$ | 0.48 | $6.069 \pm 0.007$ |
| baskball | ExDBSCAN | $1.26 \pm 0.08$ | 1.00 | $1.47 \pm 0.02$ |
| baskball | BayCon | $2.1 \pm 0.1$ | 0.82 | $1.17 \pm 0.03$ |
| blood-transfusion-service-center | ExDBSCAN | $0.67 \pm 0.09$ | 1.00 | $1.21 \pm 0.07$ |
| blood-transfusion-service-center | BayCon | $1.1 \pm 0.1$ | 0.38 | $0.7 \pm 0.4$ |
| bodyfat | ExDBSCAN | $0.9 \pm 0.2$ | 1.00 | $4.31 \pm 0.01$ |
| bodyfat | BayCon | $1.5 \pm 0.3$ | 0.36 | $3.50 \pm 0.01$ |
| breast-w | ExDBSCAN | $1.0 \pm 0.1$ | 1.00 | $3.80 \pm 0.01$ |
| breast-w | BayCon | $2.9 \pm 0.2$ | 0.82 | $3.27 \pm 0.01$ |
| chscase_census2 | ExDBSCAN | $0.9 \pm 0.1$ | 1.00 | $2.21 \pm 0.01$ |
| chscase_census2 | BayCon | $1.6 \pm 0.1$ | 0.55 | $1.81 \pm 0.01$ |
| chscase_census6 | ExDBSCAN | $1.5 \pm 0.2$ | 1.00 | $0.78 \pm 0.07$ |
| chscase_census6 | BayCon | - | 0.00 | - |
| chscase_vine1 | ExDBSCAN | $1.2 \pm 0.2$ | 1.00 | $4.55 \pm 0.09$ |
| chscase_vine1 | BayCon | $3.4 \pm 0.4$ | 0.55 | $4.11 \pm 0.01$ |
| confidence | ExDBSCAN | $2.24 \pm 0.04$ | 1.00 | $1.33 \pm 0.01$ |
| confidence | BayCon | $3.17 \pm 0.05$ | 0.35 | $1.20 \pm 0.07$ |
| diabetes | ExDBSCAN | $1.4 \pm 0.1$ | 1.00 | $4.66 \pm 0.09$ |
| diabetes | BayCon | $2.4 \pm 0.4$ | 0.42 | $2.13 \pm n.a.$ |
| diabetes_numeric | ExDBSCAN | $1.0 \pm 0.1$ | 1.00 | $0.77 \pm 0.06$ |
| diabetes_numeric | BayCon | $1.3 \pm 0.1$ | 0.72 | $0.4 \pm 0.2$ |
| diggle_table_a1 | ExDBSCAN | $1.1 \pm 0.1$ | 1.00 | $1.32 \pm 0.04$ |
| diggle_table_a1 | BayCon | $1.7 \pm 0.3$ | 0.14 | $1.14 \pm 0.02$ |
| disclosure_x_noise | ExDBSCAN | $0.6 \pm 0.1$ | 1.00 | $0.72 \pm 0.09$ |
| disclosure_x_noise | BayCon | $1.2 \pm 0.1$ | 0.75 | $0.5 \pm 0.2$ |
| ecoli | ExDBSCAN | $4.0 \pm 0.1$ | 1.00 | $5.034 \pm 0.004$ |
| ecoli | BayCon | $5.71 \pm n.a.$ | 0.29 | $1.2 \pm 0.5$ |
| glass | ExDBSCAN | $1.9 \pm 0.3$ | 1.00 | $3.954 \pm 0.007$ |
| glass | BayCon | $3.2 \pm 0.6$ | 0.45 | $3.5 \pm 0.1$ |
| hayes-roth | ExDBSCAN | $1.30 \pm 0.06$ | 1.00 | $2.79 \pm 0.02$ |
| hayes-roth | BayCon | $1.51 \pm 0.08$ | 0.81 | $1.35 \pm n.a.$ |
| heart-statlog | ExDBSCAN | $1.3 \pm 0.1$ | 1.00 | $6.014 \pm 0.008$ |
| heart-statlog | BayCon | $3.2 \pm 0.2$ | 0.70 | $5.013 \pm 0.006$ |
| iris | ExDBSCAN | $1.1 \pm 0.2$ | 1.00 | $2.847 \pm 0.005$ |
| iris | BayCon | $2.7 \pm 0.2$ | 0.75 | $1.8 \pm 0.3$ |
| liver-disorders | ExDBSCAN | $0.67 \pm 0.08$ | 1.00 | $0.59 \pm 0.06$ |
| liver-disorders | BayCon | $0.9 \pm 0.1$ | 0.19 | $0.5 \pm 0.8$ |
| longley | ExDBSCAN | $1.0 \pm 0.2$ | 1.00 | $1.12 \pm 0.02$ |
| longley | BayCon | $1.63 \pm n.a.$ | 0.16 | $1.34 \pm n.a.$ |
| machine_cpu | ExDBSCAN | $1.6 \pm 0.8$ | 1.00 | $1.0 \pm 0.1$ |
| machine_cpu | BayCon | $2 \pm 1$ | 0.62 | $0.8 \pm 0.2$ |
| mu284 | ExDBSCAN | $1.02 \pm 0.08$ | 1.00 | $1.79 \pm 0.02$ |
| mu284 | BayCon | $1.7 \pm 0.1$ | 0.52 | $1.74 \pm 0.02$ |
| no2 | ExDBSCAN | $0.9 \pm 0.2$ | 1.00 | $2.49 \pm 0.03$ |
| no2 | BayCon | $1.9 \pm 0.2$ | 0.62 | $0.01 \pm 0.04$ |
| pm10 | ExDBSCAN | $1.0 \pm 0.1$ | 1.00 | $2.892 \pm 0.007$ |
| pm10 | BayCon | $2.1 \pm 0.2$ | 0.41 | $2.26 \pm 0.01$ |
| prnn_fglass | ExDBSCAN | $2.4 \pm 0.4$ | 1.00 | $4.103 \pm 0.008$ |
| prnn_fglass | BayCon | $4.2 \pm 0.3$ | 0.67 | $4.0 \pm 0.1$ |
| rabe_131 | ExDBSCAN | $1.10 \pm 0.09$ | 1.00 | $2.28 \pm 0.02$ |
| rabe_131 | BayCon | $2.1 \pm 0.2$ | 0.88 | $2.1 \pm 0.2$ |
| sleep | ExDBSCAN | $0.9 \pm 0.2$ | 1.00 | $1.01 \pm 0.04$ |
| sleep | BayCon | $1.50 \pm 0.09$ | 0.07 | $0.78 \pm 0.03$ |
| strikes | ExDBSCAN | $1.3 \pm 0.1$ | 1.00 | $2.2 \pm 0.3$ |
| strikes | BayCon | $2.5 \pm 0.2$ | 0.43 | $1.8 \pm 0.4$ |
| vehicle | ExDBSCAN | $1.0 \pm 0.7$ | 1.00 | $2.80 \pm 0.01$ |
| vehicle | BayCon | $1.6 \pm 2$ | 0.17 | $2.98 \pm 0.04$ |
| wine | ExDBSCAN | $1.3 \pm 0.2$ | 1.00 | $6.313 \pm 0.007$ |
| wine | BayCon | $3.1 \pm 0.4$ | 0.62 | $5.474 \pm 0.008$ |

Table 3: Run-time analysis comparing ExDBSCAN with BayCon. Running times (in seconds) show ExDBSCAN's being faster than BayCon.

| Dataset | Method | Run-time ($s$) |
|---|---|---|
| autoPrice | ExDBSCAN | $1.9 \pm 0.5$ |
| autoPrice | BayCon | $17 \pm 1$ |
| baskball | ExDBSCAN | $0.010 \pm 0.001$ |
| baskball | BayCon | $10 \pm 1$ |
| blood-transfusion-service-center | ExDBSCAN | $19 \pm 3$ |
| blood-transfusion-service-center | BayCon | $10 \pm 1$ |
| bodyfat | ExDBSCAN | $1.5 \pm 0.3$ |
| bodyfat | BayCon | $20 \pm 2$ |
| breast-w | ExDBSCAN | $58 \pm 9$ |
| breast-w | BayCon | $10 \pm 1$ |
| chscase census2 | ExDBSCAN | $0.010 \pm 0.001$ |
| chscase census2 | BayCon | $15 \pm 1$ |
| chscase census6 | ExDBSCAN | $0.010 \pm 0.001$ |
| chscase census6 | BayCon | $23.3 \pm 0.1$ |
| chscase vine1 | ExDBSCAN | $0.040 \pm 0.001$ |
| chscase vine1 | BayCon | $18 \pm 1$ |
| confidence | ExDBSCAN | $0.20 \pm 0.01$ |
| confidence | BayCon | $123 \pm 2$ |
| diabetes | ExDBSCAN | $128 \pm 20$ |
| diabetes | BayCon | $16 \pm 1$ |
| diabetes numeric | ExDBSCAN | $0.010 \pm 0.001$ |
| diabetes numeric | BayCon | $1.6 \pm 0.1$ |
| diggle table a1 | ExDBSCAN | $0.010 \pm 0.001$ |
| diggle table a1 | BayCon | $19 \pm 1$ |
| disclosure x noise | ExDBSCAN | $5.0 \pm 0.8$ |
| disclosure x noise | BayCon | $89 \pm 1$ |
| ecoli | ExDBSCAN | $21 \pm 11$ |
| ecoli | BayCon | $19 \pm 3$ |
| glass | ExDBSCAN | $11 \pm 2$ |
| glass | BayCon | $15 \pm 2$ |
| hayes-roth | ExDBSCAN | $0.040 \pm 0.001$ |
| hayes-roth | BayCon | $6.4 \pm 0.7$ |
| heart-statlog | ExDBSCAN | $1.1 \pm 0.2$ |
| heart-statlog | BayCon | $14 \pm 1$ |
| iris | ExDBSCAN | $2.2 \pm 0.4$ |
| iris | BayCon | $9 \pm 1$ |
| liver-disorders | ExDBSCAN | $2.89 \pm 0.04$ |
| liver-disorders | BayCon | $6 \pm 2$ |
| longley | ExDBSCAN | $0.010 \pm 0.001$ |
| longley | BayCon | $21 \pm 1$ |
| machine cpu | ExDBSCAN | $2.1 \pm 0.3$ |
| machine cpu | BayCon | $11 \pm 2$ |
| mu284 | ExDBSCAN | $3.6 \pm 0.7$ |
| mu284 | BayCon | $17 \pm 1$ |
| no2 | ExDBSCAN | $15 \pm 2$ |
| no2 | BayCon | $12 \pm 2$ |
| pm10 | ExDBSCAN | $4.1 \pm 0.6$ |
| pm10 | BayCon | $13 \pm 1$ |
| prnn fglass | ExDBSCAN | $11 \pm 2$ |
| prnn fglass | BayCon | $14 \pm 2$ |
| rabe 131 | ExDBSCAN | $0.010 \pm 0.001$ |
| rabe 131 | BayCon | $9 \pm 1$ |
| sleep | ExDBSCAN | $0.010 \pm 0.001$ |
| sleep | BayCon | $23 \pm 1$ |
| strikes | ExDBSCAN | $0.24 \pm 0.01$ |
| strikes | BayCon | $17.6 \pm 0.2$ |
| vehicle | ExDBSCAN | $72 \pm 12$ |
| vehicle | BayCon | $29 \pm 1$ |
| wine | ExDBSCAN | $0.66 \pm 0.08$ |
| wine | BayCon | $17 \pm 2$ |

Table 4: Proximity diversity and validity for DiCE. A feed-forward neural network is fitted as a surrogate on DBSCAN's cluster labels. The neural network is differentiable allowing DiCE to be applied. Compared to ExDBSCAN's results DiCE achieves significantly worse proximity and similar diversity.

| Dataset | ProximityRand | DiversityRand |
|---|---|---|
| autoPrice | $4.8 \pm 0.3$ | $4.4 \pm 0.2$ |
| baskball | $2.3 \pm 0.1$ | $1.7 \pm 0.05$ |
| blood-transfusion-service-center | $3.6 \pm 0.2$ | $3.3 \pm 0.2$ |
| bodyfat | $4.9 \pm 0.3$ | $4.6 \pm 0.2$ |
| breast-w | $3.0 \pm 0.2$ | $1.90 \pm 0.09$ |
| chscase census2 | $5.2 \pm 0.4$ | $3.1 \pm 0.2$ |
| chscase census6 | $7.6 \pm 0.6$ | $3.5 \pm 0.3$ |
| chscase vine1 | $3.0 \pm 0.2$ | $2.4 \pm 0.1$ |
| confidence | $2.50 \pm 0.08$ | $1.6 \pm 0.2$ |
| diabetes | $4.3 \pm 0.1$ | $3.1 \pm 0.2$ |
| diabetes numeric | $1.9 \pm 0.2$ | $0.99 \pm 0.03$ |
| diggle table a1 | $2.4 \pm 0.1$ | $1.08 \pm 0.06$ |
| disclosure x noise | $1.50 \pm 0.07$ | $1.04 \pm 0.03$ |
| ecoli | $6.7 \pm 0.5$ | $5.3 \pm 0.7$ |
| glass | $4.4 \pm 0.3$ | $4.2 \pm 0.2$ |
| hayes-roth | $1.94 \pm 0.06$ | $1.22 \pm 0.05$ |
| heart-statlog | $3.7 \pm 0.2$ | $2.9 \pm 0.2$ |
| iris | $2.2 \pm 0.1$ | $2.10 \pm 0.1$ |
| liver-disorders | $2.9 \pm 0.4$ | $2.6 \pm 0.3$ |
| longley | $2.2 \pm 0.2$ | $1.7 \pm 0.2$ |
| machine cpu | $4.6 \pm 0.3$ | $3.3 \pm 0.1$ |
| mu284 | $7.5 \pm 0.2$ | $5.3 \pm 0.2$ |
| no2 | $2.7 \pm 0.2$ | $2.4 \pm 0.1$ |
| pm10 | $2.29 \pm 0.09$ | $2.52 \pm 0.05$ |
| prnnfglass | $5.2 \pm 0.3$ | $4.6 \pm 0.2$ |
| rabe 131 | $2.7 \pm 0.2$ | $2.1 \pm 0.1$ |
| sleep | $4.2 \pm 0.3$ | $2.8 \pm 0.1$ |
| strikes | $2.91 \pm 0.03$ | $1.55 \pm 0.02$ |
| vehicle | $3.7 \pm 0.5$ | $3.8 \pm 0.2$ |
| wine | $3.3 \pm 0.2$ | $3.7 \pm 0.2$ |
| wine-quality-red | $7.3 \pm 0.6$ | $5.2 \pm 0.2$ |
| wine-quality-white | $7.1 \pm 0.4$ | $4.5 \pm 0.2$ |

# C    EXTENDED BAYCON

In the experimental evaluation, comparison with BayCon was made choosing, for each instance, 10 counterfactuals for both ExDBSCAN and BayCon. Due to the nature of the method, BayCon does not allow to set a specific number of counterfactuals giving the user several alternatives, the first 10 proposed where chosen for evaluation an comparison with ExDBSCAN. For a more comprehensive analysis, we add here two ways to subset BayCon counterfactuals. Respectively, we choose 10 random options and the 10 closest to the original instance. The random choice favours diversity neglecting proximity while the proximity-oriented choice does not take into account diversity, as testified by the results in Table 5 showing inferior results to ExDBSCAN.

## C.1    RANDOM BENCHMARK

To further benchmark ExDBSCAN perform, we select 10 random core points from the target cluster and generate counterfactuals according to Eq. 4. The random benchmark does not take into account proximity but does not bias any counterfactual to be closer to a second one making it thus a good option for diversity. This intuition is confirmed by the results (Table 6) showing competitive diversitive w.r.t. to ExDBSCAN while providing worse proximity.

## C.2    QUALITATIVE EXAMPLE

To better showcase ExDBSCAN's real-world applications we designed a small qualitative example reproducing data coming from different sensors. Specifically we generate a dataset of 100 instances having 4 features: *temperature*, *pressure*, *humidity* and *voltage*. The data is organised into 3 compact normally distributes clusters. One outlier is created with temperature and pressure being out-of-distribution for every cluster. We generate counterfactuals toward each of the 3 clusters for the outlying instance which is labelled as noise by DBSCAN.

The outlying instance is $(85, 110, 70, 5)$ with features indicating temperature in Celsius, pressure in kiloPascal, percentage of humidity and voltage in Volt respectively. We generate with ExDBSCAN the initial counterfactual (maximally proximal) for each of the 3 clusters which read $(27.4, 101.6, 57.4, 5.05, )$, $(37.2, 102.9, 48.5, 5.1)$ and $(21.0, 99.8, 69.0, 4.8)$ respectively.

Analysing the proposed counterfactuals, it's clear that temperature is an outlying feature of the anomaly, as it significantly drops for all ExDBSCAN's output. Pressure as well is dropped in every counterfactual. Percentage of humidity is in line with cluster 3's counterfactuals while proposed CEs do not significantly modify voltage. Thus, the outlier wasn't assigned to cluster 1 and 2 because of temperature, pressure and voltage while it wasn't assigned to cluster 3 because of its temperature and pressure.

Table 5: Proximity diversity and validity for BayCon. Random refers to chosing the 10 counterfactuals randomly from the proposed one. Closest refers to selecting the 10 closest counterfactuals.

| Dataset | Method | Proximity | Diversity | Validity |
|---|---|---|---|---|
| autoPrice | BayCon (Random) | $3.45 \pm 0.17$ | $6.4 \pm 0.3$ | 0.46 |
| autoPrice | BayCon (Closest) | $2.67 \pm 0.14$ | $0.18 \pm 0.01$ | 0.46 |
| baskball | BayCon (Random) | $1.92 \pm 0.09$ | $1.42 \pm 0.02$ | 0.78 |
| baskball | BayCon (Closest) | $1.29 \pm 0.09$ | $0.83 \pm 0.04$ | 0.78 |
| blood-transfusion-service-center | BayCon (Random) | $1.8 \pm 0.2$ | $1.19 \pm 0.08$ | 0.55 |
| blood-transfusion-service-center | BayCon (Closest) | $1.2 \pm 0.2$ | $1.62 \pm 0.18$ | 0.55 |
| bodyfat | BayCon (Random) | $1.7 \pm 0.2$ | $3.8 \pm 0.1$ | 0.40 |
| bodyfat | BayCon (Closest) | $0.9 \pm 0.2$ | $0.29 \pm 0.01$ | 0.40 |
| breast-w | BayCon (Random) | $3.0 \pm 0.2$ | $3.6 \pm 0.1$ | 0.90 |
| breast-w | BayCon (Closest) | $2.3 \pm 0.2$ | $0.31 \pm 0.01$ | 0.90 |
| chscase census2 | BayCon (Random) | $1.6 \pm 0.1$ | $1.86 \pm 0.06$ | 0.58 |
| chscase census2 | BayCon (Closest) | $1.1 \pm 0.1$ | $0.58 \pm 0.02$ | 0.58 |
| chscase census6 | BayCon (Random) | $-$ | $-$ | 0 |
| chscase census6 | BayCon (Closest) | $-$ | $-$ | 0 |
| chscase vine1 | BayCon (Random) | $3.3 \pm 0.3$ | $4.2 \pm 0.1$ | 0.63 |
| chscase vine1 | BayCon (Closest) | $2.7 \pm 0.3$ | $0.26 \pm 0.01$ | 0.63 |
| confidence | BayCon (Random) | $3.09 \pm 0.04$ | $1.23 \pm 0.08$ | 0.35 |
| confidence | BayCon (Closest) | $2.76 \pm 0.07$ | $0.73 \pm 0.02$ | 0.35 |
| diabetes | BayCon (Random) | $3.3 \pm 0.2$ | $3.7 \pm 0.2$ | 0.42 |
| diabetes | BayCon (Closest) | $1.8 \pm 0.2$ | $0.35 \pm 0.03$ | 0.42 |
| diabetes numeric | BayCon (Random) | $1.5 \pm 0.1$ | $0.7 \pm 0.1$ | 0.72 |
| diabetes numeric | BayCon (Closest) | $1.0 \pm 0.1$ | $2.9 \pm 0.5$ | 0.72 |
| diggle table a1 | BayCon (Random) | $1.9 \pm 0.3$ | $1.5 \pm 0.1$ | 0.18 |
| diggle table a1 | BayCon (Closest) | $1.5 \pm 0.3$ | $0.68 \pm 0.05$ | 0.18 |
| disclosure x noise | BayCon (Random) | $1.4 \pm 0.1$ | $1.11 \pm 0.09$ | 0.70 |
| disclosure x noise | BayCon (Closest) | $0.8 \pm 0.1$ | $2.4 \pm 0.2$ | 0.70 |
| ecoli | BayCon (Random) | $5.90 \pm 0.08$ | $2.5 \pm 0.5$ | 0.17 |
| ecoli | BayCon (Closest) | $5.71 \pm 0.01$ | $0.6 \pm 0.2$ | 0.17 |
| glass | BayCon (Random) | $3.9 \pm 0.4$ | $4.5 \pm 0.2$ | 0.58 |
| glass | BayCon (Closest) | $3.3 \pm 0.5$ | $0.27 \pm 0.01$ | 0.58 |
| hayes-roth | BayCon (Random) | $1.75 \pm 0.06$ | $2.6 \pm 0.1$ | 0.81 |
| hayes-roth | BayCon (Closest) | $1.02 \pm 0.06$ | $0.65 \pm 0.03$ | 0.81 |
| heart-statlog | BayCon (Random) | $3.4 \pm 0.2$ | $5.4 \pm 0.1$ | 0.93 |
| heart-statlog | BayCon (Closest) | $2.6 \pm 0.2$ | $0.20 \pm 0.01$ | 0.93 |
| iris | BayCon (Random) | $2.4 \pm 0.1$ | $2.2 \pm 0.1$ | 0.80 |
| iris | BayCon (Closest) | $2.2 \pm 0.1$ | $0.50 \pm 0.02$ | 0.80 |
| liver-disorders | BayCon (Random) | $2.5 \pm 0.5$ | $1.8 \pm 0.1$ | 0.90 |
| liver-disorders | BayCon (Closest) | $1.7 \pm 0.5$ | $0.91 \pm 0.09$ | 0.90 |
| longley | BayCon (Random) | $1.7 \pm 0.2$ | $1.5 \pm 0.1$ | 0.16 |
| longley | BayCon (Closest) | $1.4 \pm 0.2$ | $0.68 \pm 0.09$ | 0.16 |
| machine cpu | BayCon (Random) | $2.2 \pm 0.2$ | $1.28 \pm 0.04$ | 0.63 |
| machine cpu | BayCon (Closest) | $1.5 \pm 0.2$ | $1.2 \pm 0.1$ | 0.63 |
| mu284 | BayCon (Random) | $1.67 \pm 0.09$ | $1.96 \pm 0.08$ | 0.48 |
| mu284 | BayCon (Closest) | $1.2 \pm 0.1$ | $0.61 \pm 0.02$ | 0.48 |
| no2 | BayCon (Random) | $1.9 \pm 0.2$ | $3.6 \pm 0.3$ | 0.59 |
| no2 | BayCon (Closest) | $1.0 \pm 0.2$ | $0.49 \pm 0.05$ | 0.59 |
| pm10 | BayCon (Random) | $2.4 \pm 0.1$ | $2.7 \pm 0.1$ | 0.61 |
| pm10 | BayCon (Closest) | $1.6 \pm 0.1$ | $0.46 \pm 0.01$ | 0.61 |
| prnn fglass | BayCon (Random) | $4.5 \pm 0.5$ | $4.7 \pm 0.3$ | 0.67 |
| prnn fglass | BayCon (Closest) | $4.0 \pm 0.6$ | $0.26 \pm 0.01$ | 0.67 |
| rabe 131 | BayCon (Random) | $2.0 \pm 0.1$ | $2.31 \pm 0.06$ | 0.85 |
| rabe 131 | BayCon (Closest) | $1.4 \pm 0.1$ | $0.49 \pm 0.02$ | 0.85 |
| sleep | BayCon (Random) | $1.9 \pm 0.6$ | $0.8 \pm 0.03$ | 0.18 |
| sleep | BayCon (Closest) | $1.8 \pm 0.6$ | $1.3 \pm 0.1$ | 0.18 |
| strikes | BayCon (Random) | $2.38 \pm 0.04$ | $2.07 \pm 0.02$ | 0.42 |
| strikes | BayCon (Closest) | $1.84 \pm 0.04$ | $0.59 \pm 0.01$ | 0.42 |
| vehicle | BayCon (Random) | $4 \pm 2$ | $3.8 \pm 0.5$ | 0.20 |
| vehicle | BayCon (Closest) | $3 \pm 2$ | $0.36 \pm 0.04$ | 0.20 |
| wine | BayCon (Random) | $2.6 \pm 0.3$ | $5.6 \pm 0.2$ | 0.55 |
| wine | BayCon (Closest) | $1.9 \pm 0.3$ | $0.20 \pm 0.01$ | 0.55 |

Table 6: Proximity diversity for the random benchmark. Specifically, 10 random core points are chosen from the DBSCAN clusters and the counterfactuals are placed $\varepsilon$ away in the direction of the original instance, analogously as the ExDBSCAN case. The random benchmark is competitive when looking at diversity, as the random choice doesn't bias any counterfactual to be closer to others. Conversely, the random benchmark doesn't perform at the same level as ExDBSCAN when looking at proximity as counterfactuals are not attracted toward the original instance.

| Dataset | ProximityRand | DiversityRand | ValidityRand |
|---|---|---|---|
| autoPrice | $4.8 \pm 0.3$ | $4.4 \pm 0.2$ | 1 |
| baskball | $2.3 \pm 0.1$ | $1.67 \pm 0.05$ | 1 |
| blood-transfusion-service-center | $3.6 \pm 0.2$ | $3.31 \pm 0.18$ | 1 |
| bodyfat | $4.9 \pm 0.3$ | $4.6 \pm 0.2$ | 1 |
| breast-w | $3.0 \pm 0.2$ | $1.90 \pm 0.09$ | 1 |
| chscase census2 | $5.2 \pm 0.4$ | $3.1 \pm 0.2$ | 1 |
| chscase census6 | $7.6 \pm 0.6$ | $3.5 \pm 0.3$ | 1 |
| chscase vine1 | $3.0 \pm 0.2$ | $2.4 \pm 0.1$ | 1 |
| confidence | $2.50 \pm 0.08$ | $1.6 \pm 0.2$ | 1 |
| diabetes | $4.3 \pm 0.1$ | $3.1 \pm 0.2$ | 1 |
| diabetes numeric | $1.9 \pm 0.2$ | $0.99 \pm 0.03$ | 1 |
| diggle table a1 | $2.4 \pm 0.1$ | $1.08 \pm 0.06$ | 1 |
| disclosure x noise | $1.49 \pm 0.07$ | $1.04 \pm 0.03$ | 1 |
| ecoli | $6.7 \pm 0.5$ | $5.3 \pm 0.7$ | 1 |
| glass | $4.4 \pm 0.3$ | $4.2 \pm 0.2$ | 1 |
| hayes-roth | $1.94 \pm 0.06$ | $1.22 \pm 0.05$ | 1 |
| heart-statlog | $3.7 \pm 0.2$ | $2.9 \pm 0.2$ | 1 |
| iris | $2.2 \pm 0.1$ | $2.1 \pm 0.1$ | 1 |
| liver-disorders | $2.9 \pm 0.4$ | $2.6 \pm 0.3$ | 1 |
| longley | $2.2 \pm 0.2$ | $1.7 \pm 0.2$ | 1 |
| machine cpu | $4.6 \pm 0.3$ | $3.3 \pm 0.1$ | 1 |
| mu284 | $7.5 \pm 0.2$ | $5.3 \pm 0.2$ | 1 |
| no2 | $2.7 \pm 0.2$ | $2.4 \pm 0.1$ | 1 |
| pm10 | $2.29 \pm 0.09$ | $2.52 \pm 0.05$ | 1 |
| prnn fglass | $5.2 \pm 0.3$ | $4.6 \pm 0.2$ | 1 |
| rabe 131 | $2.7 \pm 0.2$ | $2.1 \pm 0.1$ | 1 |
| sleep | $4.2 \pm 0.3$ | $2.8 \pm 0.1$ | 1 |
| strikes | $2.91 \pm 0.03$ | $1.55 \pm 0.02$ | 1 |
| vehicle | $3.7 \pm 0.5$ | $3.8 \pm 0.2$ | 1 |
| wine | $3.3 \pm 0.2$ | $3.7 \pm 0.2$ | 1 |
| wine-quality-red | $7.3 \pm 0.6$ | $5.2 \pm 0.2$ | 1 |
| wine-quality-white | $7.1 \pm 0.4$ | $4.5 \pm 0.2$ | 1 |

