# OpenReview forum: "ExDBSCAN: Explaining DBSCAN with Counterfactual Reasoning"
_ICLR.cc/2026/Conference — Submitted to ICLR 2026_

### Official Review · Reviewer_yYrU · 2025-10-24

**Soundness:** 2
**Presentation:** 2
**Contribution:** 2
**Rating:** 4
**Confidence:** 3

**Summary:**

The paper introduces ExDBSCAN, a method for generating counterfactual explanations tailored to the DBSCAN clustering algorithm. Unlike existing model-agnostic methods tailored for supervised learning, such as BayCon, ExDBSCAN explicitly incorporates DBSCAN’s density-connectivity structure. The authors try to formalise this through a theoretical guarantee, and evaluate the method across 30 OpenML datasets using validity, proximity, and diversity metrics. Results show higher validity and comparable or better diversity and proximity than BayCon.

**Strengths:**

The paper addresses an underexplored area in explainable machine learning, i.e., counterfactual explanations for unsupervised, density-based clustering. It proposes counterfactual reasoning for DBSCAN. The author identify a conceptual gap between existing black-box counterfactual methods and the discrete, graph-based nature of DBSCAN, proposing a theoretically grounded alternative.

In terms of quality, the paper provides formal reasoning, including proofs for validity and NP-hardness, and conducts experiments across a broad suite of datasets. The evaluation framework (validity, proximity, and diversity) is decent. However, some metrics typical CF metrics are missing (listed under limitations).

The presentation clarity is decent. The motivation is well contextualised against prior work.

Regarding significance, while the contribution is somewhat narrow, it fills a real methodological gap. The proposed approach could form a foundation for future extensions to other density-based or hierarchical clustering algorithms, and it contributes to the growing intersection of XAI and unsupervised learning.

**Weaknesses:**

The paper’s main limitations come from missing experimental and methodological detail (listed under questions). In addition, the contribution remains somewhat narrow, being tailored to a single clustering algorithm (DBSCAN) without discussing how the proposed approach could generalise or adapt to other forms of unsupervised learning.

**Questions:**

1.It would be helpful to describe in more detail how the ten BayCon counterfactuals per instance were chosen for evaluation. Since many counterfactual generators return a large set of counterfactuals, different sampling strategies may emphasise different aspects of performance. For example, selecting the highest-scoring counterfactuals might highlight quality but reduce diversity, while random sampling might empathise diversity. Clarifying this choice, or reporting both variant, would help interpret the proximity and diversity results more precisely.

2. Missing evaluation measures:
- sparsity or the number of features typically modified per counterfactual
- indication of computational cost (BayCon has several metrics related to this)
- indication of result variability (i.e., please report average values + standard deviation or some other measurement of results deviation)

3. Discussion points
- A short explanation on why the HDBSCAN-specific BayCon variant (Spagnol et al., 2024) was not included could help position ExDBSCAN more clearly relative to other density-based methods
- Can the method accommodate categorical or mixed-type features?
- Can the method explain new, unseen data points without rerunning DBSCAN? I believe this is is one of the highlights of the method proposed by (Spagnol et al., 2024)
- can the method be adapted to other forms of unsupervised learning?

---

> ### Author Response · Authors · 2025-11-21
>
> We thank the reviewer for their time and insightful feedback. As noted, ExDBSCAN is a model specific method. Its model-specific nature allows it to leverage DBSCAN’s inner workings to produce high-quality counterfactuals which is the pro of model-specific solutions (Carletti et al. Eng. Appl. Artif. Intell. 2023). The high popularity of DBSCAN makes ExDBSCAN contributions valuable to a wide population of users. All revisions are highlighted in blue throughout the updated paper.
>
> **Q1:**
>
> As you note, a downside of __BayCon__ is that it does not accept the number of counterfactuals as a parameter. In our case, we modified the implementation of the authors to stop the generation once 10 counterfactuals are reached, thus not needing any further selection strategy to find counterfactuals.
>
> **Q2:**
>
> - **Q2.1:** In the development of ExDBSCAN we prioritised validity, proximity, diversity, and actionability, as we found these properties the most critical for CEs in the DBSCAN setting. Sparsity is less natural in density-based clustering: moving a point into a dense region typically requires coordinated changes across multiple features rather than sparse, single-feature edits. Enforcing sparsity would therefore risk violating validity or pushing counterfactuals outside the target cluster. That said, ExDBSCAN does not preclude sparsity. A sparsity-aware variant could be obtained by restricting the set of allowable core points (e.g., thresholding feature-wise difference) and/or placing counterfactuals in the $\varepsilon$ neighbourhood of a core point but not necessarily $\varepsilon$ away or by post-processing counterfactuals with a sparsification step, both directions we view as promising extensions for future work.
>
> - **Q2.2** In the revised version of the paper we will provide a run time analysis showing how ExDBSCAN is more computationally efficient than BayCon in the counterfactuals search.
>
> - **Q2.3** We will report variability of the measured metrics reporting standard deviation in the Appendix of the revised version.
>
> **Q3:**
> - **Q3.1** As explained in Section 3.2 of the paper by Spagnol et al, their HDBSCAN* BayCon variant exploited probabilistic definitions specific of HDBSCAN*.  They adopt BayCon as the optimisation engine, but modify the scoring function specifically for the clustering algorithm. So for HDBSCAN*, they leverage the soft membership probabilities provided by the hierarchical density estimation. The scoring function uses HDBSCANs continuous signals that do not exist for DBSCAN.  We elaborate on DBSCAN’s inapplicability due to the missing algorithm-specific continuous scoring also in the Answer to Reviewer TwFz28 W3/Q7.
>
> - **Q3.2** Yes it can, for example by choosing a proper distance metric e.g. Gower’s. It is advisable to use the same categorical features approach used by the DBSCAN instance explained whether it is Gower’s distance or e.g. one-hot encoding.
>
> - **Q3.3** Indeed, ExDBSCAN can explain an unseen point and its position in the current fixed clustering without rerunning DBSCAN. If the idea would also be to update the clustering, one could resort to ideas similar to incremental DBSCAN (Ester et al. VLDB 1998), where only local changes in the clustering need to be taken into account, drastically limiting the work required to update the model. Model-agnostic methods like Baycon would struggle to work with incremental changes.
>
> - **Q3.4** It could be adapted to other density-based clustering e.g. hierarchical density-based OPTICS with minor tweaks.

---

> > ### Comment · Reviewer_yYrU · 2025-11-26
> >
> > I thank the authors for the clarifications. However, I remain concerned about the way BayCon was evaluated. Limiting BayCon to the first 10 generated counterfactuals effectively means the method was not given sufficient time to converge. This is analogous to stopping the training of a neural network after only a few epochs simply because it produced 10 predictions. As a result, the comparison does not reflect BayCon’s actual performance, and I am afraid this evaluation is not valid.
> >
> > Furthermore, the statement that additional metrics will be included in the revised manuscript does not provide reassurance on whether these results will meaningfully improve the evaluation. Without seeing the actual numbers, variability measures, or runtime comparisons, it is difficult to assess whether the revisions address the concerns raised. I am therefore not fully confident that the current evidence supports the claimed performance.

---

> > > ### Author Response · Authors · 2025-12-01
> > >
> > > Thank you for the follow-up and opportunity to clarify the BayCon evaluation.
> > >
> > > Regarding BayCon generation, there seems to be a misunderstanding. Of course, we allow Baycon to converge, and afterwards take the first 10 counterfactuals returned. Thus, the full BayCon method is NOT able to achieve ExDBSCAN performance.
> > >
> > > For sake of clarification of this misunderstanding, in our appendix we now include the following additional experimental results:
> > > - BayCon-Random: the first 10 BayCon samples (matching our original setup)
> > > - BayCon-Closest: the 10 BayCon samples with the smallest proximity (i.e., best BayCon outputs), independently of generation order.
> > >
> > > The random counterfactuals neglect proximity while the closest ones neglect diversity. The extended comparison shows that ExDBSCAN still achieves higher validity and diversity, while maintaining competitive proximity.

---

### Official Review · Reviewer_YsWu · 2025-10-25

**Soundness:** 2
**Presentation:** 2
**Contribution:** 2
**Rating:** 4
**Confidence:** 4

**Summary:**

This paper introduces ExDBSCAN, the first method specifically designed to generate counterfactual explanations for the DBSCAN clustering algorithm. The work addresses a significant gap in explainable AI by bringing counterfactual reasoning which is a well-established approach in supervised learning to the domain of unsupervised density-based clustering.
The paper makes three primary contributions. First, it introduces ExDBSCAN as the counterfactual explanation method tailored specifically for DBSCAN, capable of handling both noise-to-cluster and cluster-to-cluster transitions. Second, it presents a physics-inspired optimization framework that balances proximity and diversity. This framework models candidate counterfactuals as charged particles that repel each other while being connected to the original point via spring-like forces, with all distances measured through a density-connected weighted graph that respects DBSCAN's cluster structure. Third, the method provides theoretical guarantees.
However, many of the claims are overstated. They are merely natural extension of some previous works and amalgamating those work.

**Strengths:**

The paper makes three primary contributions. First, it introduces ExDBSCAN as the counterfactual explanation method tailored specifically for DBSCAN, capable of handling both noise-to-cluster and cluster-to-cluster transitions. Second, it presents a physics-inspired optimization framework that balances proximity and diversity. This framework models candidate counterfactuals as charged particles that repel each other while being connected to the original point via spring-like forces, with all distances measured through a density-connected weighted graph that respects DBSCAN's cluster structure. Third, the method provides theoretical guarantees.

**Weaknesses:**

Applying counterfactual reasoning to DBSCAN, while useful, represents an incremental extension of well-established counterfactual concepts to a new algorithm rather than a fundamentally novel approach to explainability. The contribution is narrowly focused on one specific clustering algorithm (DBSCAN), which limits its broader impact compared to more generalizable explainability frameworks.

The novelty primarily lies in combining these physics-inspired forces with the density-connected weighted graph representation. While this combination is tailored to DBSCAN, the individual components (repulsion for diversity, attraction for proximity, graph-based distances) are not novel. Graph representation is domain-appropriate but not novel: Using shortest-path distances in weighted graphs to measure similarity is standard in graph-based algorithms. The contribution here is recognizing that DBSCAN's density-connectivity should be modeled this way which is insightful but somewhat obvious given DBSCAN's definition.

Guaranteed by construction, not theory: Theorem 3.1 states that counterfactuals are valid because they're placed within ε-neighborhoods of target core points. This is essentially guaranteed by the algorithm's design (Equation 4) rather than being a non-trivial theoretical result. The "proof sketch" simply restates the assignment definition.
Circular reasoning: The paper defines the assignment function (Section 3.1) specifically to make this guarantee possible, then claims the guarantee as a contribution. This is somewhat circular—the validity is built into the definition rather than being a surprising theoretical property.
NP-hardness result (Proposition 2) is expected: Showing that the energy minimization is NP-hard connects to known results about maximum diversity problems. While formally establishing this is useful, it's an expected rather than surprising result, and the proof essentially reduces to known MDP hardness.
Proposition 1 is trivial: The observation that minimizing spring energy alone selects k-nearest neighbors is straightforward and requires minimal proof.

**Questions:**

Does the method's complete specificity to DBSCAN limit its scientific contribution? Have you considered whether your approach could extend to other density-based methods (OPTICS, DENCLUE), and if not, does this indicate the contribution is more engineering than research?

Why didn't you compare against fitting a surrogate classifier to DBSCAN's cluster assignments and then applying standard counterfactual methods? You mention this approach in Section 2 but dismiss it without empirical evaluation. Wouldn't this be the most obvious baseline?

You prove NP-hardness then use a greedy approximation. How much better is this than simply selecting k diverse points from the nearest neighbors? Have you ablated against simpler baselines like: (a) k-nearest core points, (b) farthest-first traversal on the graph, (c) random sampling from nearby core points?

You state "fixing the clustering" and defining assignment via ε-neighborhoods ensures validity, then claim guaranteed validity as a contribution. Couldn't any method achieve "perfect validity" by defining the problem to make it tautologically true? What makes your guarantee non-trivial?

**Details Of Ethics Concerns:**

No ethical concerns are identified.

---

> ### Author Response · Authors · 2025-11-21
>
> We thank you for your valuable feedback and interesting questions. We are happy to answer them in the following answers. All revisions are highlighted in blue throughout the updated paper.
>
> **W1:**
>
> As you point out, ExDBSCAN targets specifically DBSCAN. While CEs are widely researched in the context of classification methods, this is not the case for clustering methods. Spagnol et al. introduce two specific approaches tailored to k-means and HDBSCAN*, which are not applicable to DBSCAN. This highlights the need to develop an algorithm aiming at explaining the fundamental clustering approach for density-based clustering. DBSCAN's wide application renders the explainability of its output even more important. We see the model specificity as an advantage, as we benefit from knowing the inner workings of the methods, taking advantage of them to produce a better explanation (Carletti et al. Eng. Appl. Artif. Intell. 2023). Still, we also see the possibility to extend our work to other density-based methods in future works, this would enhance their interpretability by proposing proximal and diverse counterfactuals.
>
>
> **W2:**
>
> ExDBSCAN is the first method that provides high-quality counterfactual explanations tailored specifically to DBSCAN. We believe the combination of the physics-inspired objective, the core-point graph formulation, and the DBSCAN-specific constraints constitutes a novel contribution in that they are chosen to reflect the density-based notion and create counterfactuals that do not e.g. simply use proximity alone.
>
> Many influential XAI methods build upon established ideas while innovating through how these components are adapted and integrated. For example, SHAP applies Shapley values, an established concept, to machine learning using an approximation strategy, yet the resulting framework is widely recognised as novel and impactful. Our intention is similar: to bring together existing principles in a way that enables counterfactual reasoning in a setting where no such method previously existed.
>
> **W3 / Q4:**
>
> We agree that our generated counterfactuals are valid by construction, which testifies to our solution approach, rather than being shortcoming. Please note that the concept of validity is strictly based on the existing definition for classification methods (Eq. 1) extended to the clustering task. The cluster assignment (lines 206-209) is not altered in our work, it is a reformulation of DBSCAN's definitions of core, border, and noise points.
>
> The counterfactual generation method, placing the counterfactual in the epsilon neighborhood of a core-point (Eq. 4), is our model-specific design choice to ensure validity. This allows us to guarantee that the CEs we identify are valid by construction while optimising for proximity and diversity within the set of valid counterfactual explanations.
> Regarding proofs, we leverage them to establish core properties and it is not so much their (limited) complexity that is in focus.

---

> > ### Author Response · Authors · 2025-11-21
> >
> > **Q1:**
> >
> > Model-specificity allows a method to better tune to an algorithm’s inner-workings. The more the method is widely applied, the more research on it is impactful. Consider explanation methods like Grad-CAM, they produce heatmaps explaining why an image was classified as it was. The fact that it is not model-agnostic and is instead applied to neural networks doesn’t downgrade it as engineering, especially given the prominent use of neural networks. Another example is DIFFI (Carletti et al. Eng. Appl. Artif. Intell. 2023) which explains the highly popular isolation forest for anomaly detection. In our opinion, works like DIFFI constitute valuable research output. Similarly, DBSCAN is a widely recognised clustering method that lay the foundation for density-based clustering as a paradigm. It is used in a variety of application domains, where explanations are expected to be of high impact.
> >
> > That being said, ExDBSCAN could potentially  be adapted to other density-based methods like HDBSCAN and OPTICS. Some changes may be advised e.g. based on the specific assignment definition, but the framework of a greedy algorithm that balances repulsion and attraction can be used unchanged.
> >
> > **Q2:**
> >
> > We agree that fitting a surrogate to DBSCAN labels and subsequently evaluating classical CE generation methods seems like the straightforward baseline. We have chosen to not include them due to noise handling, surrogate complexity and lack of fidelity guarantees. DBSCAN denotes not only cluster but also non-cluster points, i.e., noise points, that do not lie in dense areas. The lack of meaningful decision boundaries for this “class” makes it hard for any surrogate classifier to learn the highly irregular or disconnected regions. Additionally when training a surrogate classifier, more design decision, such as architecture, hyperparameters, or tuning steps, highly influence the quality and outcome of the surrogate model. As a consequence, the comparison between the original DBSCAN clustering and classifiers outcome is less controlled and thus less interpretable. Even if the surrogate learns a good approximation, we cannot ensure that what the model learned aligns to DBSCAN’ density-reachability concept. While CEs would explain the surrogates decisions, this is not directly transferable to DBSCAN. We will include a more detailed discussion of surrogate approaches in the revised version of the paper. We recognize that this presents an intriguing experimental setup to examine potential differences in the information extracted by a classifier using DBSCAN labels compared to DBSCAN and will add experiments like it until the end of the discussion phase
> >
> > **Q3:**
> >
> > Picking the k-nearest neighbours would forfeit diversity and account for proximity alone. Furthest-first traversal would instead be a solid choice for diversity while proposing counterfactuals that are on the opposite part of the cluster with respect to the closest to the original point. We will add a random selection of points as a benchmark in a revised version of the paper.

---

### Official Review · Reviewer_EGW5 · 2025-10-28

**Soundness:** 3
**Presentation:** 3
**Contribution:** 2
**Rating:** 4
**Confidence:** 5

**Summary:**

The paper introduces ExDBSCAN, a method for generating counterfactual explanations for DBSCAN clustering. ExDBSCAN directly exploits DBSCAN’s density-based structure to produce valid, proximal, and diverse counterfactuals. Given a factual point and a target cluster the method produces a set of k diverse counterfactuals belonging to that cluster. To achieve this, it determines k core points of the target cluster that are mutually distant (in DBSCANs connectivity graph) but close to the factual. Core point selection is based on greedy optimization of a cost function. For each core point a counterfactual is placed in its \epsilon-region on the line segment connecting the core point and the factual.

**Strengths:**

S1. It is the first attempt to compute counterfactuals in the context DBSCAN clustering.
S2. The method produces valid counterfactuals (belonging to the target cluster).
S3. It is interesting that two different types of distance functions are used.
S4. The paper is well-written and easy to read.
S5. Experimental results using several datasets are presented.

**Weaknesses:**

W1. The proposed objective formulation is typical, when multiple CFEs are needed: a diversity term plus a proximity term.  There are two major simplifications: i) the search is restricted to the set of core points and ii) minimization is performed in a greedy manner. Technical depth is rather limited. In the case of k=1 (one CFE produced), the approach is trivial.
W2. Although several datasets are considered, experimental comparison is insufficient: I strongly suggest to compare with an indirect approach where a surrogate classifier is build using the cluster labels and then CFEs are computed using one of the several available libraries for explaining classifiers.
W3. In my opinion, the use of the natural metaphor does not add value to the presentation of the method. Also propositions and theorems seem to be trivial.

**Questions:**

Q1. See comment W2 above. The comparison with Baycon (which is agnostic) does not add value to the paper.
Q2. Why is it difficult to formulate the method considering the core points of all possible target clusters?
Q3. Since k CFEs are generated independently for each cluster, how the CFE sets obtained for several target clusters could be combined into a final set with k CFEs?
Q4. The way that non-actionable features are handled should be better explained in the Appendix.
Q5. Is it possible to find the optimal solution for k=1 (one CFE)?

---

> ### Author Response · Authors · 2025-11-21
>
> We appreciate the reviewer for the constructive comments and acknowledging the novelty, clarity, and experimental breadth of our work. All updates are highlighted in blue in the revision.
>
> **W1:**
>
> Our two design choices are motivated by the following reasons:
>
> - **Restriction to core points.** Core points define a DBSCAN cluster: every cluster consists of all points that lie within epsilon of at least one core point. Therefore, the epsilon-neighbourhoods of core points cover the entire region that DBSCAN recognises as belonging to the cluster. Since a counterfactual must fall inside this region to be valid, searching within core-point epsilon neighbourhoods is not an arbitrary restriction, but the natural and correct search space for DBSCAN counterfactuals.
>
> - **Greedy optimisation.** Proposition 2 shows that the optimisation problem is NP-hard, so an exact solution is not infeasible and an approximation is required. Our greedy strategy performs well empirically. We agree that the k = 1 case is simple, but this does not weaken the method; in fact, k = 1 is the only setting where the greedy algorithm is guaranteed to find the optimal solution to Eq. 3. The contribution of ExDBSCAN lies in producing multiple, diverse counterfactuals, something that requires a principled selection strategy, not a trivial extension. Without such a strategy, the method risks selecting core points that are redundant (low diversity) or far from the instance (low proximity), resulting in counterfactual sets that are uninformative or implausible. Our greedy criterion explicitly avoids these issues balancing proximity and diversity at each step.
>
> **W2/Q1:**
>
> Surrogate-based comparisons are conceptually interesting, and we appreciate the reviewer’s suggestion. In the revised version of the paper we will add surrogate models as a benchmark. However, in the context of DBSCAN, surrogate-based counterfactual generation introduces several challenges:
> - **Noise handling.** DBSCAN’s noise points do not form a coherent or learnable “class,” and a surrogate would need to approximate highly irregular or disconnected regions, leading to poor fidelity.
>
> - **Surrogate complexity.** A surrogate adds its own uncertainty through extra hyperparameters, tuning, and optimisation, making comparisons less controlled. Even a highly accurate surrogate offers no guarantee of reproducing DBSCAN’s density-reachability rules. Mislearned boundaries would yield counterfactuals valid only for the surrogate, not DBSCAN, meaning such CEs explain the surrogate rather than the clustering model and are therefore not suitable for evaluation.
>
> - **Instability concerns.** Prior work (Visani et al., JORS 2022) shows that surrogates trained on synthetic or held-out samples can be unstable, especially with complex or non-convex decision boundaries such as those arising in density-based clustering.
> Most classification-oriented CE methods require differentiable models or class probabilities, which DBSCAN does not provide. We therefore use __BayCon__, the only model-agnostic baseline that works solely from discrete cluster assignments. The clustering-specific method mentioned by the reviewer (Spagnol et al. arXiv 2022)adapts __BayCon__ only for $k$-means and HDBSCAN, where continuous scoring functions exist; such scoring functions are not available for DBSCAN, making __BayCon__ the only practical comparator.
>
> **W3:**
>
> Assuming the comment refers to the physics-inspired formulation: our intention was algorithmic rather than metaphorical. Electrostatic- and spring-like potentials offer a principled, interpretable way to combine repulsion (diversity) and attraction (proximity). These well-understood optimisation templates help avoid arbitrary weighting schemes and give intuitive insight into the solution. If this was not the aspect intended, please let us know so we can address it directly.
>
> **Q2/Q3:**
>
> ExDBSCAN can indeed be extended to operate without specifying a target cluster. This requires only a minor modification: the repulsion (Coulomb-like) term is computed only within each cluster, since points in different clusters are already distinct by assignment. This variant can then select diverse core points across all clusters simultaneously, producing a unified set of $k$ counterfactuals without requiring the user to choose a specific destination cluster. We have clarified this extension in the main paper and expanded the discussion in Appendix A.2.3 .
>
> **Q4:**
>
> Appendix A.5 offers an explanation along with a supporting figure. We would be grateful if you could point out anything unclear so that we can refine further.
>
> **Q5:**
>
> Yes, for $k = 1$, ExDBSCAN simply selects the core point that minimises the spring-like term, i.e., the closest core point in the relevant cluster. This is the global optimal solution to Eq. $3$ for $k = 1$, and indeed the only setting where the greedy algorithm is provably optimal. We now clarify further this property in the paper.

---

### Official Review · Reviewer_TwFz · 2025-10-28

**Soundness:** 1
**Presentation:** 1
**Contribution:** 2
**Rating:** 2
**Confidence:** 4

**Summary:**

The paper proposes ExDBSCAN, a counterfactual‑explanation framework tailored to DBSCAN. The method constructs a weighted graph over core points, then formulates an energy function that balances proximity to the instance being explained via a spring‑like attraction, and diversity among counterfactuals via electrostatic repulsion. The method maximizes weighted graph distance and minimizes l2 distance between points in a train dataset. The authors prove that every generated counterfactual is valid (i.e., lies in the target cluster) and that the underlying optimisation problem is NP‑hard. A greedy approximation algorithm is used, and extensive experiments on 30 OpenML tabular datasets show that ExDBSCAN attains perfect validity, lower proximity, and higher diversity than the model‑agnostic Bayesian baseline BayCon. The paper also discusses handling non‑actionable features and presents theoretical arguments for the optimisation hardness.

**Strengths:**

## Novelty
First method that guarantees validity for counterfactuals in a density‑based clustering setting.

## Theoretical grounding
Proof of validity, NP‑hardness, and a clear energy formulation that connects to physics.

## Empirical evaluation
Large benchmark (30 datasets), three metrics, clear visualisation.

## Practical relevance
Addresses a real interpretability gap for DBSCAN, which is widely used in many domains.

## Reproducibility
Code and data links provided; experiments are reproducible from the description.

## Clarity of motivation
The paper convincingly explains why existing CE methods fail for DBSCAN and motivates the physics‑inspired approach.

**Weaknesses:**

## Algorithmic description
The greedy procedure is only sketched. The counterfactual sampling algorithm is unclear. It would be better to provide pseudocode for the algorithm.

## Complexity / Scalability
Building the full $\epsilon$‑neighbourhood graph is $O(n^2)$ in worst case. No discussion of practical runtimes, memory usage, or scalability to high‑dimensional / large datasets.

## Baseline selection
Only BayCon is used. Other clustering‑aware CE methods [1,2] could provide a more diverse comparison. I am also interested in the comparison with the regular CE methods like DiCE.
[1] Zhou, Peng, et al. "EACE: Explain Anomaly via Counterfactual Explanations." Pattern Recognition 164 (2025): 111532.
[2] Spagnol, Aurora, et al. "Counterfactual Explanations for Clustering Models." arXiv preprint arXiv:2409.12632 (2024).

## Existence of counterfactuals
The paper guarantees validity of produced CEs but does not discuss what happens when no feasible counterfactual exists.

## Metric choice
Proximity is measured by Euclidean distance to the explained point, while DBSCAN may use other metrics. This metric might not be the best for categorical features. No sensitivity analysis to feature scaling or metric choice.

## Non‑actionable features
The handling of non‑actionable attributes is only briefly described; real‑world constraints (categorical features, monotonicity, bounds) are not addressed.

## Human evaluation
The metrics (validity, proximity, diversity) are quantitative but do not directly capture interpretability or usefulness to end‑users. This work would benefit from some example explanations provided by the proposed method and user-study.

## Statistical analysis
No significance tests are reported. The bar‑plot differences may be due to random sampling of points.

## Documentation of proofs
The NP‑hardness proof is sketched; a full formal argument would strengthen the theoretical contribution.

**Questions:**

1. Could you provide pseudocode or a detailed description of the greedy algorithm? How many iterations does it run, and what is its time complexity in terms of n (number of points) and k (desired number of CEs)?

2. Have you benchmarked ExDBSCAN on larger datasets (e.g., > 50k points)? What is the memory footprint of storing the weighted graph, and can you use approximate nearest‑neighbour techniques?

3. DBSCAN is agnostic to the distance metric; does ExDBSCAN require Euclidean? If a different metric (e.g., cosine) is used, how would the proximity and diversity terms change?

4. In cases where no actionable feature can change the cluster assignment, how does ExDBSCAN behave? Does it return an empty set or a “failed” flag?

5. Your non‑actionable feature treatment filters core points based on Euclidean distance in a subspace. How would you handle discrete features or domain constraints that are not Euclidean?

6. Have you conducted any user studies to confirm that the generated counterfactuals are perceived as actionable or informative? If not, could you discuss plans for such evaluation?

7. Why was Spagnol et al. 2024 or EACE not included? Would they provide a useful benchmark, especially for outlier‑related counterfactuals?

8. How sensitive are the counterfactuals to $\epsilon$ and $minPts$? If a suboptimal parameter set is used, do you still obtain valid counterfactuals?

---

> ### Author Response · Authors · 2025-11-21
>
> We thank the reviewer for the detailed and insightful feedback. We appreciate the recognition of novelty, theoretical soundness, and empirical thoroughness of our work, and we hope the reviewers concerns are addressed sufficiently. All revisions are highlighted in blue throughout the updated paper.
>
> **W1/Q1:**
>
> Thank you for the suggestion! We agree that providing pseudocode will improve clarity of the description in Section 3.2. In the revised version, we provide the complete pseudocode in Appendix A.2, with greedy core-point selection procedure and the mapping from the selected core points to counterfactual points.
>
> Moreover, we will also provide details of the complexity. With $m$ being the number of core points in the target cluster and $k$ being the number of counterfactuals requested, at each greedy step, we evaluate the change in energy for at most $m$ candidates. Since this evaluation is constant time after precomputing the graph distances, the greedy selection has the time complexity of: $O(km)$ - also elaborated in the appendix A.2.
>
> **W2/Q2:**
>
> We thank the reviewer for raising this point. The worst-case time complexity for the graph is $O(n^2)$ with memory being $O(n^2)$ as well. We would like to point out that the graph construction is “for free” with the neighbourhood information obtained from running  DBSCAN which the user can store in practical applications. We will expand Section 3.2 to explicitly discuss practical runtime considerations and moreover add to the appendix indepth description of the graph construction cost and memory footprint.
>
> **W3/Q7:**
>
> Our experimental evaluation uses __BayCon__ because it can generate counterfactuals for clustering without requiring differentiability or probabilistic outputs; both of which DBSCAN lacks. The DBSCAN assignment function is discrete and does not provide probabilities or gradients meaning that:
> __DiCE__ and similar CE methods are not applicable, as they rely on continuous scores and often on gradient or probability shifts to guide the optimisation. DiCE is therefore unable to produce CE for DBSCAN for this reason.
> __EACE__ is specifically designed for outlier detection methods, while we focus on complete DBSCAN clusterings rather than just the outliers / noise points. EACE’s definition requires a continuous anomaly score and optimises this score to move anomalies into inliers, which is not available in DBSCAN. Additionally, EACE does not accept target clusters nor support multiple destination clusters. It relies on outliers being rare, analogous to DBSCAN noise points, but this does not generalise to cluster-to-cluster transitions. For these reasons EACE cannot be directly compared to ExDBSCAN.
> Spagnol et al. also adopt __BayCon__ as the optimisation engine, but modify the scoring function specifically for the clustering algorithm. So for k-means, they rely on distances to centroids and for HDBSCAN, they leverage the soft membership probabilities provided by the hierarchical density estimation. Both scoring functions use algorithm-specific continuous signals that do not exist for DBSCAN.
>
> Since __BayCon__ is the only model-agnostic CE method that operates using only the discrete cluster assignment, it is the only existing  baseline for fair comparison with ExDBSCAN. We will highlight this more clearly in the revision of the paper and discuss in more detail the limitations of adapting other methods as outlined above.
>
> **W4/Q4:**
>
> Thanks for addressing this important point. When all features are actionable, a counterfactual always exists, because a trivial valid counterfactual is simply any core point of the target cluster.
> When some features are non-actionable, we constrain the set of eligible core points, as described in Appendix A.9.2. If this constrained set is empty, i.e., no core point's epsilon neighbourhood is reachable without modifying non-actionable features, then no feasible counterfactual exists, i.e., the set of valid counterfactuals is empty. Raising a flag or returning an empty set would thus be equivalent as the only case in which a counterfactual is not proposed is when none exists. When counterfactuals exist, ExDBSCAN has provably a validity of 1.

---

> > ### Author Response · Authors · 2025-11-21
> >
> > **W5/Q3/Q5:**
> > - **Metric choice.**
> >   - ExDBSCAN is, by design, metric-agnostic, exactly like DBSCAN. It can be run without modification using any metric that is used to fit DBSCAN, e.g., Minkowski, cosine, Mahalanobis, or mix-typed metrics. We can replace the metric in our algorithm in any step (reference core point selection, CE assignment, etc.) with the one chosen for the DBSCAN clustering. We highly recommend using the same distance metric as DBSCAN, ensuring that ExDBSCAN’s notion of: density reachability, graph connectivity, proximity and diversity are fully consistent with the underlying changes.
> >   - Moreover, for categorical or mixed-type attributes, DBSCAN (and thereby ExDBSCAN) can be run using appropriate mixed metrics such as Gower distance (Gower biometrics 1971), which naturally extends to the weighted graph and shortest-path distances used by our method.
> >
> >
> > - **Non-actionable features.**
> >   - When a feature is non-actionable, we restrict accepted core points to those whose epsilon-neighbourhood is reachable without modifying that feature. Using a mixed-type metric, categorical differences contribute a non-zero distance; therefore, any core point whose categorical value differs from that of the original point will lie farther than epsilon in the non-actionable subspace and will be automatically excluded. This filtering mechanism naturally enforces categorical, discrete, or other domain constraints (e.g., monotonicity or bounds) without modifying the algorithm, i.e., the constraints are incorporated simply by excluding core points that violate them.
> >   - We note in the paper other metrics and refer to Appendix A.4 in the updated version for details.
> >
> > **W6:**
> >
> > In the revised version of the paper, we will expand Appendix A.5, which addresses non-actionable features. For categorical variables, selecting an appropriate metric, as already discussed in the previous response, enables ExDBSCAN to function effectively with categorical attributes. Additionally, monotonicity and bounds can be easily incorporated through further filtering of the core point set. Instead of restricting the set of reference core points to actionable dimensions only, we can additionally restrict this to be a specific direction or range depending on the constraints defined by the user. This allows us to reduce the set of reference core-points to not only generate actionable CEs in the sense of selected features but also actionable regarding real-world constraints.
> >
> > **W7/Q6:**
> >
> > We agree that qualitative or user-centred evaluation would be beneficial, but have not yet conducted a human-subject study. Counterfactuals are widely regarded in the social sciences as an especially effective and intuitive explanation form (e.g., (Miller Artif. Intell. 2019)), which partially motivates their adoption here.
> >
> > To complement the quantitative evaluation, we will add a small qualitative example in the appendix illustrating the interpretive value of ExDBSCAN counterfactual sets in a real dataset. We also include conducting a full user study as an interesting direction for future work
> >
> > **W8:**
> >
> > To reduce the effect of random sampling, we perform multiple runs and present the mean across these runs. Performing statistical tests poses challenges since we are unable to compare the results between individual datasets regarding proximity or diversity, as these values are specific to each dataset. In the revised version, we include the standard deviation in Tables 1 and 2, to highlight that the differences illustrated in the original version of the paper do not stem from random sampling.
> >
> > **W9:**
> >
> > A complete version of the NP-hardness argument is provided in Appendix A.5. It reduces the energy minimisation problem to a variant of the maximum diversity problem, which is known to be NP-hard and then relies on a proof-by-contradiction step. We rely on an established hardness result.
> >
> > **Q8:**
> >
> > Experiments were performed across a range of datasets and DBSCAN parameters ($\varepsilon$ ranges from 0.3 to 3 and MinPts from 3 to 30). Across the diverse datasets tested, ExDBSCAN generated valid CEs and performed robustly across these parameter ranges.
> >
> > Importantly, validity is invariant to parameter choice, since ExDBSCAN only uses DBSCAN’s fixed cluster structure, and our approach guarantees to only generate valid CEs (if they exist). Suboptimal parameters may influence cluster shapes, but ExDBSCAN will still produce valid counterfactuals for the resulting clustering. In practice, this can even be used diagnostically to identify poorly chosen DBSCAN parameters. If noise points are very close to clusters, i.e., low proximity (proximity close to $\varepsilon$ ), this might be an indicator for a poorly chosen $\varepsilon$ value.

---

> > > ### Comment · Reviewer_TwFz · 2025-11-26
> > >
> > > Thank you for your detailed response to my review. I appreciate the effort to address the concerns raised.
> > > # On Algorithmic Description (W1/Q1)
> > > I acknowledge the addition of pseudocode in Appendix A.2 and the complexity analysis.
> > >
> > > # On Scalability (W2/Q2)
> > > Your response that graph construction is "for free" because it reuses DBSCAN's neighborhood information is somewhat misleading. While true in principle, this assumes users store this information, which is not standard practice. More importantly, you have not provided any empirical runtime analysis or demonstrated scalability on larger datasets. The claim that you will "expand Section 3.2" and "add to the appendix" suggests these details were missing from the original submission, which is concerning for a methods paper.
> > >
> > > # On Baseline Selection (W3/Q7)
> > > I find your justification for excluding other baselines partially convincing but incomplete. While I accept that DiCE requires gradients and EACE targets outlier detection specifically, possible evaluation might be using surrogate classifier trained on DBSCAN clusters.  A single baseline makes it difficult to contextualize your method's performance. Even comparing against a simple nearest-neighbor-in-target-cluster baseline would strengthen the evaluation.
> > >
> > > # On Metric Choice and Non-Actionable Features (W5/Q3/Q5)
> > > Your claim that ExDBSCAN is "metric-agnostic" is helpful, but all experiments use Euclidean distance. Without empirical validation using other metrics (cosine, Gower, etc.), this remains a theoretical claim. The suggestion to use Gower distance for categorical features is reasonable but untested.
> > >
> > > # On Statistical Analysis (W8)
> > > Adding standard deviations is a step forward, but proper statistical testing (e.g., Wilcoxon signed-rank tests across datasets) remains absent. Showing mean ± std does not establish statistical significance of the observed differences.
> > >
> > > # Summary
> > > While the authors have made good-faith efforts to address my concerns, several key issues remain:
> > >
> > > - No empirical scalability analysis
> > > - Limited baseline comparison
> > > - No experimental validation of claimed metric-agnosticism
> > > - No user study or meaningful qualitative evaluation
> > >
> > > I am willing to raise my score given the improvements in algorithmic description and the commitment to include standard deviations, but the paper still falls short of the standards required for acceptance at this venue.

---

> > > > ### Author Response · Authors · 2025-12-01
> > > >
> > > > **On Scalability (W2/Q2)**
> > > >
> > > > Thank you for returning to this point. As you also noted, the focus of our work is on proposing a new physics-inspired explanation approach that handles density-based clustering notions. While more detailed scalability analysis could further characterise our work, we find it important to note that by logging already computed information in DBSCAN, no additional computational effort is needed to construct the graph, which we consider a major computational advantage.
> > > >
> > > > We add additional empirical runtime analysis to comply with your request. The results confirm that ExDBSCAN shows clearly superior performance over BayCon. Moreover, we note that ExDBSCAN operates only on the core points of the target cluster, not on the full dataset. Thus, its complexity grows gracefully with cluster size, rather than dataset size. This is a key distinction that we will highlight more clearly. These explanations only support, but do not add new information to the fact that ExDBSCAN remains practically efficient even for larger datasets.
> > > >
> > > >
> > > > **On Baseline Selection (W3/Q7)**
> > > >
> > > > ExDBSCAN makes a novel contribution in an area which has received relatively limited attention thus far, which also means there are no readily available baselines beyond BayCon that add to the characterisation of ExDBSCAN and the problem it solves. While a nearest-neighbour-in-target-cluster baseline may serve the purpose of testing a naive solution, it is bound to suffer from the inability to incorporate any diversity nor structural information that are the two properties at the core of generating counterfactual explanations in our setting. As a consequence, it would be expected to perform poorly on the diversity metric, and to be highly sensitive to local fluctuations. Nevertheless, we have added a random baseline, in which core points from the target cluster are chosen randomly,  in the revised version of our work to demonstrate the performance gain of ExDBSCAN.
> > > >
> > > > Regarding surrogate classifiers, a fair comparison is particularly challenging. This is due to the fact that any classifier would make different assumptions that may deviate from DBSCANs notion, in particular regarding the separation of noise from clusters. Surrogate classifiers are thus biased by the choice of classifier, with issues such as instability, and lack of fidelity guarantees to DBSCAN’s density structure, as well as hyperparameter and tuning overhead (Visani et al., JORS 2022). As a result, it is not clear that any particular surrogate classifier would be representative or a fair competitor. In the revised version of our paper, we fit a feed-forward neural network allowing DiCE to produce counterfactuals. Compared to ExDBSCAN, diversity results are competitive while in terms of proximity DiCE produces significantly worse results than ExDBSCAN.
> > > >
> > > > **On Metric Choice and Non-Actionable Features (W5/Q3/Q5)**
> > > >
> > > > We make two clarifications in the revision:
> > > >
> > > > 1. Metric-agnosticism refers to the fact that ExDBSCAN uses the same metric as DBSCAN for both attraction and repulsion terms; thus, any extension to other metrics is inherited directly from DBSCAN.
> > > >
> > > > 2. While we agree that an evaluation with multiple diverse metrics (and categorical variables) is interesting and a clear direction for future work, we note that BayCon, however, only utilises Gower’s distance and does not allow usage of other metrics. We focus our paper on numerical features due to the complexity of the mixed-typed data scenario in a density-based setting. When working with density-based applications, a fair weighting of the continuous and categorical part is challenging, with Gower’s distance biasing the definition of density toward categorical variables.
> > > >
> > > > **On Statistical Analysis (W8)**
> > > >
> > > > Thank you for the suggestion, we will also add a critical-difference plot, including the nearest-neighbour baseline to provide a clearer ranking-based comparison across datasets to confirm the stability of our findings.
> > > >
> > > > **Note on Summary**
> > > >
> > > > We sincerely appreciate the constructive comments, but we consider our work complete in its presentation of the ExDBSCAN problem, conceptual and algorithmic solution, as well as community standard evaluation metrics. We will include the above outlined revisions to further improve our work. Extension to new data types or application domains, and user studies are considered interesting future work.

---

### Meta-Review · Area_Chair_8hoP · 2026-01-03

**Summary:**

The paper introduces ExDBSCAN, a counterfactual explanation method for DBSCAN. While the motivation is clear, the submission falls short in several critical areas:

* Algorithmic depth: The greedy algorithm remains only superficially described; for $𝑘=1$ it is trivial.

* Scalability: No convincing empirical runtime or large‑scale analysis; claims of efficiency are theoretical.

* Baselines: Evaluation relies almost entirely on BayCon; surrogate models, DiCE, EACE, or simple baselines are missing or inadequately tested. While DiCE was added in the rebuttal, other requested baselines are still missing

* Metric choice: Claimed metric‑agnosticism is unvalidated empirically (all experiments use Euclidean distance) and not properly contextualized leading to misunderstandings.

* Statistical rigor: No significance testing.

* Human evaluation: No user study or qualitative evidence of interpretability.

Despite rebuttal efforts, these core issues remain unresolved. The contribution is incremental, with insufficient empirical and practical validation.

**Reviewer Concerns:**

Reviewer TwFz (Score: 2 → likely 4)

Acknowledged pseudocode and runtime clarifications, but still found scalability, baselines, and metric validation lacking. Would raise slightly but remain below threshold.

Reviewer EGW5 (Score: 4 → remain 4)

The rebuttal added clarifications on core points and greedy optimization. However, the concerns on technical novelty and baselines remain unadressed.

Reviewer YsWu (Score: 4 → remain 4)

The reviewer had concerns on the technical novelty, evaluation breadth, lack of ablations, and the overall impact of the work. The discussion in the rebuttal was somewhat subjective and I do not see that changing the original concerns of the reviewers. The authors did not perform any expriment to address the concers in a quantitative and factual manner.

Reviewer yYrU (Score: 4 → remain 4)

BayCon evaluation still judged invalid; runtime and variability measures not convincing. Score unchanged.

**Reviewer Scores:**

I do not see any of the reviewers changing their scores to a positive rating since the core issues remain unresolved despite the rebuttal. A summary of the unresolved concerns are already discussed above.

---

### Decision · Program_Chairs · 2026-01-26

Reject